# FEW-SHOT ATTRIBUTE LEARNING

## ABSTRACT

Semantic concepts are frequently defined by combinations of underlying attributes. As mappings from attributes to classes are often simple, attribute-based representations facilitate novel concept learning with zero or few examples. A significant limitation of existing attribute-based learning paradigms, such as zero-shot learning, is that the attributes are assumed to be known and fixed. In this work we study the rapid learning of attributes that were not previously labeled. Compared to standard few-shot learning of semantic classes, in which novel classes may be defined by attributes that were relevant at training time, learning new attributes imposes a stiffer challenge. We found that supervised learning with training attributes does not generalize well to new test attributes, whereas self-supervised pre-training brings significant improvement. We further experimented with random splits of the attribute space and found that predictability of test attributes provides an informative estimate of a model's generalization ability.

## 1 INTRODUCTION

Formation of class concepts is one of the most fundamental processes in machine perception. Although class concepts are often defined based on their attribute information, e.g., *birds* are warm-blooded vertebrates that lay eggs and have feathers, attributes are rarely considered in a typical machine perception system that directly maps from input signals to output classes. Humans also leverage similarity in the attribute space to recognize classes, which are "*information-rich bundles of attributes that form natural discontinuities*" (Rosch & Mervis, 1975). The acquisition of attribute knowledge therefore helps us build a more compact and efficient representation that is useful for the perception of classes.

Another distinct advantage of using attribute information is that it facilitates learning new classes with few or even zero examples, which has been leveraged in studies of zero-shot learning (ZSL) (Palatucci et al., 2009; Lampert et al., 2014; Misra et al., 2017). Other models use attributes as direct outputs before the classes for improved modularity and interpretability (Farhadi et al., 2009; Koh et al., 2020). However, all of these attribute-based models rely on a pre-defined set of attributes, that are shared among all classes. Consider a learning agent deployed in the wild. Although the agent may learn new classes by composing some of the existing attributes, its learning capability would be greatly improved if it can expand its attribute vocabulary.

Motivated by this learning scenario, we are interested in the problem of learning new attributes that are previously not labeled in the dataset. This is a step towards continual learning (Van de Ven & Tolias, 2019), where the reward function evolves and the system must adapt if the reward becomes dependent on previously irrelevant input attributes. In particular, we focus on a few-shot learning (FSL) setup (Lake et al., 2011; Vinyals et al., 2016), where only a few positive and negative examples of the target attributes are available, to model the rapid adaptation task.

Considering few-shot attribute learning has an extra benefit, in that it can provide manipulable factors to study generalization. In standard few-shot learning, object semantic classes are split into training and test; however, there is still a lack of understanding of when models transfer their knowledge from training classes to test ones. Since semantic classes can often be defined with a set of attributes, a split in the attribute space therefore provides us a finer control on the degree to which training classes are *related* to test ones. By studying the transfer performance on novel attributes, we expect our work can generate insight into the generalization performance on semantic classes in standard few-shot learning.

**Celeb-A**

**Zappos-50K**

Figure 1: **Sample FSAL episodes using Celeb-A (left) and Zappos-50K (right).** Positive and negative examples are sampled according to attributes.

To study this challenging task of few-shot attribute learning (FSAL), we contribute new benchmark datasets consisting of images of faces (Celeb-A) (Liu et al., 2015), shoes (Zappos50K) (Yu & Grauman, 2014), and general objects (ImageNet-with-Attributes) (Deng et al., 2009). Unlike in standard few-shot learning where supervised pre-training generally helps learning, surprisingly we found that directly supervising the model with a set of training attributes does not generalize well on the test attributes, whereas self-supervised pre-training brings significant improvement. We further ran experiments with random splits of the attribute space and discovered that the predictability of attributes provides an informative estimate of a model's ability to generalize. The few-shot attribute learning paradigm proposed in this paper will facilitate more efficient and flexible continual learning and shed a light on the practical understanding on generalization of novel concepts.

## 2 FEW-SHOT ATTRIBUTE LEARNING

In this section, we define our few-shot attribute learning (FSAL) paradigm and highlight the additional challenges of FSAL compared to the standard few-shot learning of semantic classes.

Similar to standard few-shot learning, at test time, the learner is presented with an episode of data. The support set consists of $N$ positive and negative examples of the target attributes

$$\mathcal{S} = \{(\mathbf{x}_1^{S+}, 1), \ldots, (\mathbf{x}_N^{S+}, 1), (\mathbf{x}_1^{S-}, 0), \ldots, (\mathbf{x}_N^{S-}, 0)\}, \tag{1}$$

where the $+$ or $-$ superscript suffix denotes whether the input is a positive or negative example. After rapid learning on the support set, the model is then evaluated on the binary classification performance of the query set:

$$\mathcal{Q} = \{(\mathbf{x}_1^{Q+}, 1), \ldots, (\mathbf{x}_M^{Q+}, 1), (\mathbf{x}_1^{Q-}, 0), \ldots, (\mathbf{x}_M^{Q-}, 0)\}. \tag{2}$$

At test time, the target binary label may concern a novel attribute that was previously unlabeled in the training set. For example, in one test episode, a smiling face with *eyeglasses* is positively labeled alongside other faces with eyeglasses. The task here is to learn the attribute of "wearing eyeglasses". However, while the learner might have seen training images with eyeglasses, it was never a relevant feature for the purpose of predicting the positively labeled instances.

Furthermore, suppose that in another test episode, the same *smiling* face is positively labeled alongside other smiling faces. The target attribute here has now changed from "wearing eyeglasses" to "smiling." This highlights a critical difference between few-shot attribute learning and standard few-shot learning of semantic classes: in standard FSL, each instance can belong to only one class regardless of the episode. In FSAL, due to the multi-label nature of the attribute space, one instance could have different labels depending on the context of the support set examples. Furthermore, there may be a large amount of ambiguity when the support set is small. Figure 1 shows a few examples of our attribute learning episodes. Note that in order to create task diversity, we allow both unary and binary attributes, where binary attributes are conjunctions of two unary attributes.

In order to solve the FSAL task, the learner must correctly determine the context. Just like in zero-shot learning, one natural way to solve this problem would be to learn to predict the underlying attributes of each image. Given the attributes, you could then estimate the context in each episode (Lampert et al., 2014). However, methods that accurately predict attributes relevant to training episodes may not generalize well, since at test time FSAL introduces novel attributes. Instead, we explore methods that allow more general representations to be learned.

## 3 METHODOLOGY

In this section, we present our proposed approach to tackle the problem of few-shot attribute learning. Two classes of approaches are frequently considered in standard few-shot learning. First,

*episodic* approaches train methods directly from a set of few-shot episodes. This class of methods can be naturally applied to our learning setting. Second, *pre-training* approaches train a network to directly classify a set of training classes, and at test time, the embedding network is transferred to solve the test task by training another classifier on top. If absolute attribute IDs are provided to the learner, then one natural approach is to instead train an attribute classifier. After the attribute classifier network is learned, we can then transfer the representations to recognize test attributes.

We explored both of these approaches in our experiments, but we found that they tend to learn good representations only for the training attributes but not for test ones. To address the generalization issues of these standard approaches, we propose a new algorithm that preserves more general features.

Our proposed method follows the pre-training paradigm and contains three stages. The first stage is pre-training the representation network using an unsupervised (contrastive) learning objective. The second stage is fine-tuning using supervision based either on attribute or episode information. Lastly, we learn a sparse linear classifier at test-time to solve a new episode. We describe each stage of learning below.

## 3.1 STAGE I: UNSUPERVISED REPRESENTATION PRE-TRAINING

Our proposed few-shot attribute learning tasks require a strongly generalizable learner. We hypothesized that learning general-purpose features that capture varying aspects of objects would be helpful to enable this desired capability. We therefore considered unsupervised representation learning, and we hypothesize that it can learn general semantic features. We chose SimCLR (Chen et al., 2020) as a representative from this category due to its empirical success. In general, contrastive learning approaches aim to build invariant representations between a pair of inputs $\{\mathbf{x}, \mathbf{x}'\}$ that are produced by applying random data augmentations (e.g. cropping) to an input image. It is likely to preserve more general semantic features since all image attributes are useful towards identifying another random crop of the same image.

We first obtain the embedding output $\mathbf{h}$ from the CNN, and then following SimCLR, we project $\mathbf{h}$ to $\mathbf{z}$ using a multi-layered perceptron (MLP): $\mathbf{h} = \mathrm{CNN}(\mathbf{x}), \mathbf{z} = \mathrm{MLP}_1(\mathbf{h})$. With a batch of image pairs denoted by $\{\mathbf{x}_i\}, \{\mathbf{x}'_i\}$, we can obtain their features $\{\mathbf{z}_i\}, \{\mathbf{z}'_i\}$, and the contrastive loss function is defined similar to the cross entropy function:

$$\mathcal{L}_1 = -\sum_i \log \frac{\exp(\mathbf{z}_i \cdot \mathbf{z}'_i/\tau)}{\sum_j \exp(\mathbf{z}_i \cdot \mathbf{z}'_j/\tau)}, \tag{3}$$

where $\tau$ is an extra temperature hyperparameter.

## 3.2 STAGE II: REPRESENTATION FINE-TUNING

In the second stage, we aim to utilize some labels from the training set to supervise the network. We consider using two different modes of supervision: 1) the FSAL binary episodic labels, or 2) the underlying binary attribute IDs. To prevent overwriting the representations and making them overly sensitive to training attributes, we add another projection MLP that learns more specific representations for finetuning on training attributes: $\mathbf{g} = \mathrm{MLP}_2(\mathbf{h})$. The fine-tuning objectives we consider are as follows.

- Unsupervised-then-FineTune-on-Episodes (**UFTE**). We adopt the Prototypical Networks (Snell et al., 2017) formulation, where the network solves a learning episode of $N$ positive and negative support examples by using prototypes $\mathbf{p}$: $\mathbf{p}^+ = \frac{1}{N}\sum_i \mathbf{g}_i^+; \mathbf{p}^- = \frac{1}{N}\sum_i \mathbf{g}_i^-$. With query example $\mathbf{g}^q$, we can make a binary prediction:

$$\hat{y}^q = \frac{\exp(-d(\mathbf{g}^q, \mathbf{p}^+))}{\exp(-d(\mathbf{g}^q, \mathbf{p}^+)) + \exp(-d(\mathbf{g}^q, \mathbf{p}^-))}, \tag{4}$$

where $d$ is some dissimilarity score, e.g. Euclidean distance or cosine dissimilarity, and the training objective is to minimize the classification loss between the prediction $\hat{y}^q$ and the label $y^q$:

$$\mathcal{L}_{2E} = \sum_j -y_j \log \hat{y}_j^q - (1 - y_j^q) \log(1 - \hat{y}_j^q), \tag{5}$$

where $j$ is the index of query examples.

| Paradigm | Test time task | Task specification |
|---|---|---|
| ZSL (Lampert et al., 2014) | Novel semantic classes of existing attributes | Attribute IDs |
| CZSL (Misra et al., 2017) | Novel combinations of existing attributes & classes | Attribute IDs |
| FSL (Lake et al., 2011) | Novel semantic classes | Support examples |
| FSAL (Ours) | Novel (previously unlabeled) attributes | Support examples |

Table 1: **Differences between zero-shot learning (ZSL), compositional ZSL (CZSL), few-shot learning (FSL), and our newly proposed few-shot attribute learning (FSAL).** Our task requires the model to generalize to new attributes.

- Unsupervised-then-FineTune-on-Attributes (**UFTA**). With persistent attribute information, we can train a linear classifier with sigmoid activation to directly predict the absolute attribute labels $\mathbf{a}$: $\hat{\mathbf{a}} = W_A \mathbf{g} + b_A$, with the loss being

$$\mathcal{L}_{2A} = \sum_k -\mathbf{a}_k \log \hat{\mathbf{a}}_k - (1 - \mathbf{a}_k) \log(1 - \hat{\mathbf{a}}_k), \qquad (6)$$

where $k$ is the index of attributes.

### 3.3 STAGE III: FEW-SHOT LEARNING

Once representations are learned, it remains to be decided how to use the small support set of each given test episode in order to make predictions for the associated query set. We consider three alternative approaches. MatchingNet (Vinyals et al., 2016) uses a nearest neighbor (**NN**) classifier, whereas ProtoNet (Snell et al., 2017) uses the nearest centroid (**NC**). Following Chen et al. (2019), we propose to directly learn a logistic regression (**LR**) on top of the representation. This approach learns a weight coefficient for each feature dimension, thus performing some level of feature selection, unlike the NC or NN variants. Still, the weights need to be properly regularized to encourage high-fidelity selection. This is important since each episode only focuses on a new classification criteria using a subset of features. For this, we apply an L1 regularizer on the weights to encourage sparsity. In this way, the learning of a classifier is essentially done at the same time as the selection of feature dimensions. The overall objective of the classifier is:

$$\arg\min_{\mathbf{w}, b} -y \log(\hat{y}) - (1 - y) \log(1 - \hat{y}) + \lambda \|\mathbf{w}\|_1, \qquad (7)$$

where $\hat{y} = \sigma(\mathbf{w}^\top \mathbf{h} + b)$, and $\mathbf{h}$ is the representation vector extracted from the CNN backbone. Note that in this stage we discard the projection MLPs that are defined in previous stages since they are trained towards training attributes and we found that they do not transfer well to novel attributes.

## 4 RELATED WORK

**Few-shot learning:** Few-shot learning (FSL) (Fei-Fei et al., 2006; Lake et al., 2011; Vinyals et al., 2016) entails learning new tasks with only a few examples. With an abundance of training data, FSL is closely related to the general meta-learning or learning to learn paradigm (Thrun, 1998), as a few-shot learning algorithm can be developed on training tasks and run on novel tasks at test time. In standard few-shot classification, each image only has a single unambiguous class label, whereas in our few-shot attribute learning, the target attributes can vary depending on how the support set is presented. We show in this paper that this is a more challenging problem as it requires the model to be more flexible and generalizable. In early benchmarks, a set of semantic classes was randomly split into a training and test set. We hypothesize that this often leads to a common set of attributes that span (most of) the training and test classes, thus causing high transferability between these two sets, which allows simple solutions based on feature re-use (Chen et al., 2019; Raghu et al., 2020) to work well. Later benchmarks explicitly attempt to vary the separation between train and test classes, based on varying the distances in the underlying WordNet classes (*tiered*-ImageNet (Ren et al., 2018)), or in different image domains (Meta-Dataset (Triantafillou et al., 2020)). However, we argue that reasoning about the underlying attributes directly offers a more systematic framework to measure the relatedness and transferability between the train and test set. We expect our analysis to open the door to such studies in the future. Few-shot attribute learning is also related to multi-label few-shot learning (Alfassy et al., 2019; Li et al., 2021). These prior works emphasize on the compositional aspect, whereas we propose models that address the transferability of the learned representations. Additionally, Xiang et al. (2019) explored combining incremental few-shot learning and attribute learning for pedestrian images.

**Attribute learning:** In the past, there have been a number of works that aim to predict attribute information from raw inputs Ferrari & Zisserman (2007); Farhadi et al. (2009; 2010); Wang &

| | Sup. | Celeb-A | | Zappos-50K | |
|---|---|---|---|---|---|
| | | 5-shot | 20-shot | 5-shot | 20-shot |
| Chance | - | $50.00_{\pm 0.00}$ | $50.00_{\pm 0.00}$ | $50.00_{\pm 0.00}$ | $50.00_{\pm 0.00}$ |
| MatchingNet | E | $68.30_{\pm 0.76}$ | $71.73_{\pm 0.52}$ | $77.26_{\pm 0.60}$ | $80.47_{\pm 0.49}$ |
| MAML/ANIL | E | $71.24_{\pm 0.74}$ | $73.35_{\pm 0.53}$ | $77.05_{\pm 0.50}$ | $81.10_{\pm 0.43}$ |
| TAFENet | E | $69.10_{\pm 0.76}$ | $72.11_{\pm 0.54}$ | $79.20_{\pm 0.57}$ | $83.42_{\pm 0.44}$ |
| ProtoNet | E | $72.12_{\pm 0.75}$ | $75.27_{\pm 0.51}$ | $77.22_{\pm 0.51}$ | $83.42_{\pm 0.41}$ |
| TADAM | E | $73.54_{\pm 0.70}$ | $76.06_{\pm 0.53}$ | $81.45_{\pm 0.50}$ | $86.23_{\pm 0.40}$ |
| ID | C | $69.95_{\pm 0.69}$ | $77.53_{\pm 0.53}$ | - | - |
| U | - | $73.47_{\pm 0.68}$ | $79.97_{\pm 0.51}$ | $83.88_{\pm 0.44}$ | $90.92_{\pm 0.32}$ |
| UFTE (Ours) | E | $\underline{76.69}_{\pm 0.69}$ | $\underline{82.83}_{\pm 0.48}$ | $\mathbf{85.50}_{\pm 0.42}$ | $\mathbf{92.20}_{\pm 0.28}$ |
| SA | A | $72.91_{\pm 0.74}$ | $78.86_{\pm 0.48}$ | $82.17_{\pm 0.48}$ | $88.24_{\pm 0.37}$ |
| UFTA (Ours) | A | $\mathbf{78.98}_{\pm 0.69}$ | $\mathbf{84.14}_{\pm 0.48}$ | $\underline{84.61}_{\pm 0.43}$ | $\underline{91.66}_{\pm 0.29}$ |
| **Oracles** | | | | | |
| SA* | A | $84.74_{\pm 0.60}$ | $89.15_{\pm 0.38}$ | $88.11_{\pm 0.39}$ | $93.00_{\pm 0.28}$ |
| GT | - | $91.07_{\pm 0.49}$ | $98.16_{\pm 0.17}$ | $97.66_{\pm 0.16}$ | $99.84_{\pm 0.04}$ |

Table 2: **5- and 20-shot attribute learning results on Celeb-A and Zappos-50K.** We compare standard FSL methods to variants of our approach. Representation learning based on class identity or attribute labels are outperformed by unsupervised methods that fine-tune using attribute or episode labels. Methods can be supervised by 1) *"E"*=episode binary labels, 2) *"A"*=attributes, and 3) *"C"*=face identity. The best is **bolded** and the second best is underlined.

Mori (2010). A related model is later proposed by Koh et al. (2020) to achieve better causal interpretability. There have also been a number of datasets that have been collected with visual attributes annotated (Liu et al., 2015; Yu & Grauman, 2014; Welinder et al., 2010; Patterson & Hays, 2016; Pham et al., 2021). One key difference between our work and these attribute learning approaches is that at test time we aim to learn a classifier on novel attributes that are previously not labeled in the training set, and this brings additional challenges of transfer learning and learning with limited labeled data.

**Zero-shot learning:** In zero-shot learning (ZSL) (Farhadi et al., 2009; Akata et al., 2013; Xian et al., 2019; Lampert et al., 2014; Romera-Paredes & Torr, 2015; Akata et al., 2015), a model is asked to recognize classes not present in the training set, supervised only by some auxiliary description (Ba et al., 2015) or attribute values (Farhadi et al., 2009) (see Wang et al. (2019a) for a survey). Lampert et al. (2014) studied the *direct attribute prediction* method, similar to the Supervised Attribute baseline described in Section 5.2. Compositional ZSL aims at learning classes (Misra et al., 2017; Purushwalkam et al., 2019; Wang et al., 2019b; Yang et al., 2020) defined by a novel composition of labeled attributes and object classes. An important distinction between ZSL and our few-shot attribute learning task is that ZSL uses the same set of attributes for both training and testing; by contrast, our task asks the model to learn attributes for which there are no labels during training, and they may not be relevant to any of the training attributes or episodes. We summarize the relationships between ZSL, FSL and our task in Table 1.

**Generalization to novel tasks:** One key component of our work is an attempt to understand the generalization behavior of learning novel concepts at test time. Relevant theoretical studies consider novel task generalization, casting it in a transfer learning and learning to learn framework (Baxter, 2000; Ben-David & Borbely, 2008; Ben-David et al., 2010; Pentina & Lampert, 2014; Amit & Meir, 2018; Lucas et al., 2021). A common theme in these studies is in characterizing task relatedness, and the role that it plays in generalization to novel tasks. Arnold & Sha (2021) studied task clustering for few-shot learning in the embedding space and found class splits that are of different difficulty levels. Sariyildiz et al. (2021) use the WordNet hierarchy to compute semantic distances. In our paper, we instead split the data in the attribute space, and if we assume that semantic classes are combinations of attributes, then a disjoint attribute split will imply further semantic distances. In our work, we investigate the role of task relatedness empirically by investigating generalization performance under different splits of the attribute space.

## 5 EXPERIMENTS

In this section, we evaluate our proposed UFTE and UFTA approaches on several FSAL tasks.

### 5.1 DATASETS

We consider the following three datasets:

- **Celeb-A** (Liu et al., 2015) contains over 200K images of celebrities' faces. Each image is annotated with binary attributes, detailing hair color, facial expressions, and other descriptors. We split 14 attributes for training and 13 for test.

| | Celeb-A | | | Zappos-50K | | |
|---|---|---|---|---|---|---|
| | NN | NC | LR | NN | NC | LR |
| Meta | 71.73±0.52 | 75.27±0.51 | 73.38±0.53 | 80.47±0.49 | 83.42±0.41 | 81.10±0.43 |
| U | 75.72±0.48 | 77.78±0.52 | 79.97±0.51 | 85.17±0.40 | 88.63±0.37 | 90.92±0.32 |
| UFTE | 79.03±0.47 | 81.04±0.47 | 82.83±0.48 | 86.23±0.34 | 90.61±0.31 | **92.20**±0.28 |
| SA | 75.33±0.47 | 77.24±0.51 | 78.86±0.48 | 81.17±0.44 | 85.48±0.41 | 88.24±0.37 |
| UFTA | 77.30±0.52 | 82.16±0.46 | **84.14**±0.48 | 86.40±0.36 | 90.25±0.33 | **91.66**±0.29 |
| SA* | 78.84±0.41 | 84.61±0.42 | 89.15±0.38 | 87.54±0.33 | 90.97±0.31 | 93.00±0.28 |

Table 3: **Combination of different representation & few-shot learners on 20-shot attribute learning.** Our proposed representation learning plus a simple logistic regression (LR) is consistently the best. Note: Meta-NN = MatchingNet, Meta-NC = ProtoNet, Meta-LR = MAML/ANIL.

| | Celeb-A | | | Zappos-50K | | |
|---|---|---|---|---|---|---|
| | Train attr | Test attr | Gap | Train attr | Test attr | Gap |
| ProtoNet | 87.12±0.40 | 75.09±0.52 | −12.03 | 92.88±0.24 | 83.42±0.41 | −9.46 |
| U | 79.48±0.54 | 79.97±0.51 | −0.49 | 94.03±0.23 | 90.92±0.32 | −3.11 |
| UFTE | 87.25±0.40 | 82.83±0.48 | −4.42 | 95.91±0.18 | 92.20±0.28 | −3.71 |
| SA | 88.25±0.38 | 78.86±0.48 | −9.39 | 95.11±0.19 | 88.24±0.37 | −6.87 |
| UFTA | 85.53±0.43 | 84.14±0.48 | −1.39 | 94.61±0.21 | 91.66±0.29 | −2.95 |
| SA* | 87.88±0.39 | 89.15±0.38 | +1.27 | 95.59±0.18 | 93.00±0.28 | −2.58 |

| L | D? | UFTE | | UFTA | |
|---|---|---|---|---|---|
| | | Val Acc. (Δ) | Gap | Val Acc. (Δ) | Gap |
| 0 | | 78.02 (−2.19) | −9.72 | 82.81 (+2.60) | −4.80 |
| 1 | | 76.86 (−3.35) | −11.14 | 79.56 (−0.65) | −7.43 |
| 1 | ✓ | 82.01 (+1.80) | −5.63 | 83.39 (+3.18) | −2.05 |
| 2 | | 76.32 (−3.89) | −11.58 | 79.71 (−0.50) | −7.23 |
| 2 | ✓ | 82.43 (+2.22) | −4.83 | 83.86 (+3.65) | −1.90 |

Table 4: **Comparison of representation learning methods with respect to their ability to predict training and testing attributes.** Standard methods such as ProtoNet and SA perform well on training attributes but do not transfer well to novel ones (large training vs. test gaps in red).

Table 5: **Number of projection layers (L) during finetuning, and whether they are discarded (D) during testing.** Numbers are from Celeb-A 20-shot. Δ denotes changes compared to no finetuning.

- **Zappos-50K** (Yu & Grauman, 2014) contains just under 50K images of shoes annotated with attribute values. We split these into 40 attribute values for training, and 39 for testing.
- **ImageNet-with-Attributes** is a small subset of the ImageNet dataset (Deng et al., 2009) with attribute annotations. It has 9.6K images. We used 11 attributes for training and 10 for testing.

In all of the datasets above, there is no overlap between training and test attributes. Additional split details can be found in the supplementary materials.

**Episode construction:** For each episode, we randomly select one or two attributes and look for positive examples belonging to these attributes simultaneously. We also sample an equal number of negative examples that don't match the selected attributes. This will construct a *support set* of positive and negative samples, and then we repeat the same process for the corresponding *query set* as well. Sample episodes are shown in Figure 1.

## 5.2 METHODS FOR COMPARISON

We first consider a set of classic few-shot learning methods for comparison. These methods are directly trained on FSAL episodes of training attributes.

- **MatchingNet** (Vinyals et al., 2016) is a soft version of 1-nearest-neighbor. At test time, it will retrieve the label of the support example that is the closest in the feature space.
- **MAML** (Finn et al., 2017) performs several gradient descent steps in an episode and learns the parameter initialization. For simplicity, we used the ANIL variant (Raghu et al., 2020) that only learns the last layer in the inner loop.
- **ProtoNet** (Snell et al., 2017) computes an average "prototype" for each class and retrieves the closest one.
- **TAFENet** (Wang et al., 2019c) learns a meta-network that can output task-conditioned classifier parameters.
- **TADAM** (Oreshkin et al., 2018) predicts the batch normalization parameters by using the average features of the episode. For our task we found that conditioning on the positive examples only works the best.

In addition to the approaches above, we consider the following representation learning baselines for comparison.

- **ID** trains a network to perform the auxiliary task of face identity classification (Celeb-A only).
- **SA**, or supervised attribute, resembles the "Baseline" approach in the FSL literature (Chen et al., 2019). The network learns representations by predicting the training attributes associated with each image.
- **U** denotes unsupervised representation learning (SimCLR). We train separate models on the Celeb-A and Zappos datasets. For ImageNet, we utilize the off-the-shelf model checkpoint trained on the full ImageNet-1K. Compared to our approach, this baseline skips the finetuning stage (Stage II).

| | $X$ | $A$ | ImageNet-with-Attributes | |
| --- | --- | --- | --- | --- |
| | | | 5-shot | 20-shot |
| Chance | | | $50.00 \pm 0.00$ | $50.00 \pm 0.00$ |
| MAML | | | $57.90 \pm 0.75$ | $57.46 \pm 0.70$ |
| U | ✓ | | $69.05 \pm 0.65$ | $71.25 \pm 0.62$ |
| UFTE (Ours) | ✓ | | $\underline{70.92} \pm 0.69$ | $\underline{72.12} \pm 0.63$ |
| SA | | ✓ | $64.36 \pm 0.68$ | $64.16 \pm 0.65$ |
| UFTA (Ours) | ✓ | ✓ | $\mathbf{71.12} \pm 0.65$ | $\mathbf{72.91} \pm 0.63$ |

Table 6: **5- and 20-shot attribute learning results on ImageNet.** Learners uses logistic regression (LR) at test time.

| | NN | NC | LR |
| --- | --- | --- | --- |
| Meta | $61.28 \pm 0.62$ | $61.50 \pm 0.70$ | $57.46 \pm 0.70$ |
| U | $69.63 \pm 0.59$ | $71.12 \pm 0.66$ | $71.25 \pm 0.62$ |
| UFTE | $\underline{69.77} \pm 0.57$ | $\mathbf{72.94} \pm 0.61$ | $\underline{72.12} \pm 0.63$ |
| SA | $62.42 \pm 0.62$ | $62.84 \pm 0.68$ | $64.16 \pm 0.65$ |
| UFTA | $\mathbf{71.55} \pm 0.61$ | $\underline{72.42} \pm 0.63$ | $\mathbf{72.91} \pm 0.63$ |
| SA* | $68.36 \pm 0.60$ | $70.48 \pm 0.66$ | $70.92 \pm 0.64$ |

Table 7: **20-shot FSAL on ImageNet** with different few-shot learners.

| | Train attr | Test attr | Gap |
| --- | --- | --- | --- |
| MAML | $68.16 \pm 0.59$ | $57.46 \pm 0.70$ | -10.70 |
| U | $76.36 \pm 0.60$ | $71.25 \pm 0.62$ | -5.11 |
| UFTE | $78.31 \pm 0.56$ | $\underline{72.12} \pm 0.63$ | -6.19 |
| SA | $69.03 \pm 0.66$ | $64.16 \pm 0.65$ | -4.87 |
| UFTA | $\underline{77.08} \pm 0.62$ | $\mathbf{72.91} \pm 0.63$ | -4.17 |
| SA* | $68.72 \pm 0.64$ | $70.92 \pm 0.64$ | 2.20 |

Table 8: **Training vs. test attributes** of 20-shot FSAL on ImageNet.

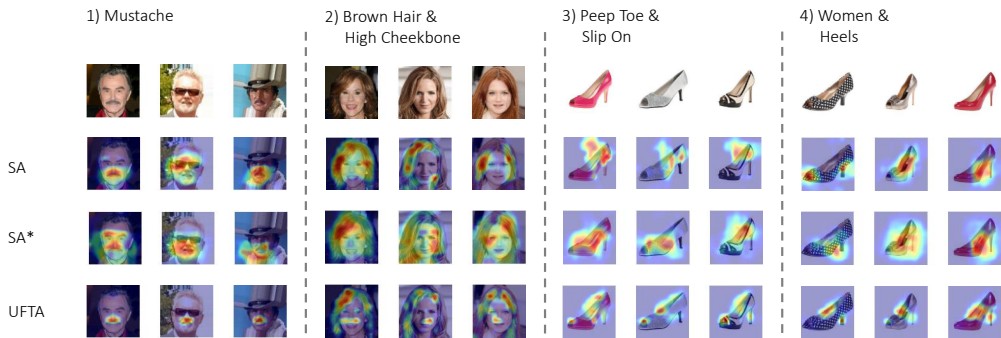

Figure 2: **Visualization of few-shot classifiers using CAM (**Zhou et al., 2016**), on top of different representations.** Left: Celeb-A; Right: Zappos-50K. Target attributes that define the episode are shown above and images are from the query set of the positive class at test time.

We also provide two oracle approaches to study the upper bound to generalization to novel attributes.

- **Oracle SA\*** learns its representations by predicting all binary attributes including both training and test ones.
- **Oracle GT** directly uses the ground-truth binary attribute values as input features, and the readout is performed by training a logistic regression. It still needs to select the active attributes that are used in each episode.

For representation learning baselines, we mainly use logistic regression (LR) in few-shot episodes, but we also report results using the nearest neighbor (NN) and nearest centroid (NC) classifiers.

**Implementation details:** For Celeb-A and Zappos, images were cropped and resized to 84×84. We used ResNet-12 (He et al., 2016; Oreshkin et al., 2018) as the CNN backbone. The projection MLPs have 512-512-128 units. We train SimCLR entirely on Celeb-A/Zappos images, i.e. *not* using pre-trained ImageNet checkpoints for fair comparison. For ImageNet-with-Attributes, we utilize the off-the-self SimCLR model from ImageNet-1k, which has access to more unlabeled images. The image dimensions are 224×224. We include additional experimental details in the Appendix.

## 5.3  MAIN RESULTS

Table 2 shows our main results on Celeb-A and Zappos-50K with 5- and 20-shot episodes. Table 3 explores different combinations of representations and few-shot learners. Overall, the standard episodic meta-learners performed relatively poorly. Also, supervised attribute (SA) learning and learning via the auxiliary task of class facial identification (ID) were not helpful for representation learning either. Interestingly, U attained relatively better test performance, suggesting that the training objective in contrastive learning indeed preserves more general features—not just for semantic classification tasks as shown in prior work, but also for the flexibly-defined attribute classes in our FSAL paradigm.

Our proposed UFTA and UFTE approaches obtained significant gains in performance, suggesting that a combination of unsupervised features with some supervised information is indeed beneficial for this task. Lastly, our methods are able to reduce the generalization gap between SA and the oracle SA*, in fact almost closing it entirely on Zappos-50K.

Results on ImageNet-with-Attributes are reported separately for clarity, because U, UFTE, and UFTA had access to additional unlabeled examples. As shown in Table 6, both UFTE and UFTA outperformed other methods substantially. Because of the additional unlabeled data available in this setting, even U achieved a substantially better accuracy than SA and MAML. Results in Table 7 show that UFTE and UFTA work well when combined with different few-shot learners.

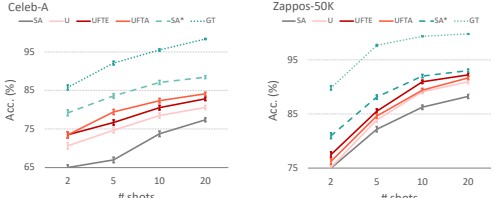 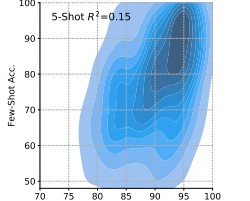 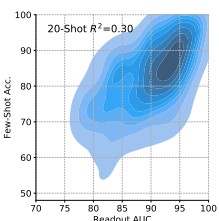

(a) **How many examples are needed for FSAL?** Performance increases with number of shots, even when given the binary ground-truth attribute vector (**GT**), suggesting that there is greater natural ambiguity in the task than in standard FSL.

(b) **Correlation between readout accuracy and few-shot accuracy using UFTA**. Part of the variance can be explained by the challenge of predicting attributes, and the rest from the ambiguity of FSAL. More shots reduce variance and improve correlation.

Figure 3: **Task ambiguity and the effect of number of shots in FSAL**

**Visualizing few-shot classifiers:** To understand and interpret the decision made by few-shot linear classifiers, we visualize the classifier weights by using CAM (Zhou et al., 2016), and plot the heatmap over the $11 \times 11$ spatial feature map in Figure 2. SA sometimes shows incorrect localization as it is not trained to classify those novel test attributes. SA* shows bigger but less precise heatmaps since the training objective encourages the propagation of attribute information spatially. In contrast, our proposed method produces surprisingly accurate and localized heatmaps that pinpoint the location of the attributes (e.g. mustache or cheekbone); this is impressive since no labeled information concerning these attributes was available during representation learning. This result supports the hypothesis that local features can be good descriptors that match different views of the same instance during contrastive learning, and finetuning further establishes a positive transfer between training and test attributes.

**Number of shots and task ambiguity:** Since we have a flexible target attribute class in each episode, it could be the case that the support examples are ambiguous. For example, by presenting only a smiling face with eyeglasses in the support set, it is unclear whether the positive set is determined by "smiling" or "wearing eyeglasses". Figure 3a show several approaches evaluated using LR with varying numbers of support examples per class in Celeb-A and Zappos-50K episodes, respectively. The oracle GT gradually approached 100% accuracy as the number of shots approached 20. This demonstrates that FSAL tasks potentially require more support examples than standard FSL to resolve ambiguity. Again here, UFTA and UFTE consistently outperformed U, SA, and ID baselines across different number of shots. Figure 3b shows the correlation between readout performance of attributes and few-shot learning accuracy, using UFTA. With a larger number of shots, there is a higher correlation between the two, but there is still a large amount of variance that is due to the ambiguity of the task itself. More details are included in the Appendix.

**Ablation studies:** Table 5 studies the effect of the projection MLP for attribute classification finetuning. Adding MLP projection layers was found to be beneficial for unsupervised learning in prior work (Chen et al., 2020). Here we found that adding MLP layers is also critical in our representation finetuning stage as well. Finetuning directly on the backbone (depth=0), and keeping the MLP during test (Discard=no) both led to significant drop in performance. In the Appendix, we also report on studies of the effect of adding the L1 regularizer on LR.

### 5.4 ANALYSIS ON FEW-SHOT GENERALIZATION

In Tables 4 and 8, we study the performance gap between training attributes and test attributes. Notably, SA performs very well on test episodes defined using training attributes, but there is a large generalization gap between training and test attributes. Our methods show significant improvements in terms of reducing the generalization gap between training and test attributes. Moreover, we find that self-supervised pre-training generally preserves informative features and is more general than supervised pre-training.

**Investigating the cause of generalization issues:** We hypothesize that the weak performance of episodic learners and SA on our benchmarks is because their training objectives essentially encourage ignoring attributes that are not useful at training time, but may still be useful at test time. In Appendix G, we study a synthetic problem to further analyze these generalization issues. We explore training a ProtoNet model on data from a linear generative model, where each FSAL episode presents ambiguity in identifying the relevant attributes. In this setting, the network is forced to discard information that is useful for test tasks in order to solve the training tasks effectively, and thus fails to generalize.

**Transferability score:** Up to this point, we have only studied one particular split of training/test attributes. We would like to examine whether our conclusions generalize over different splits. More

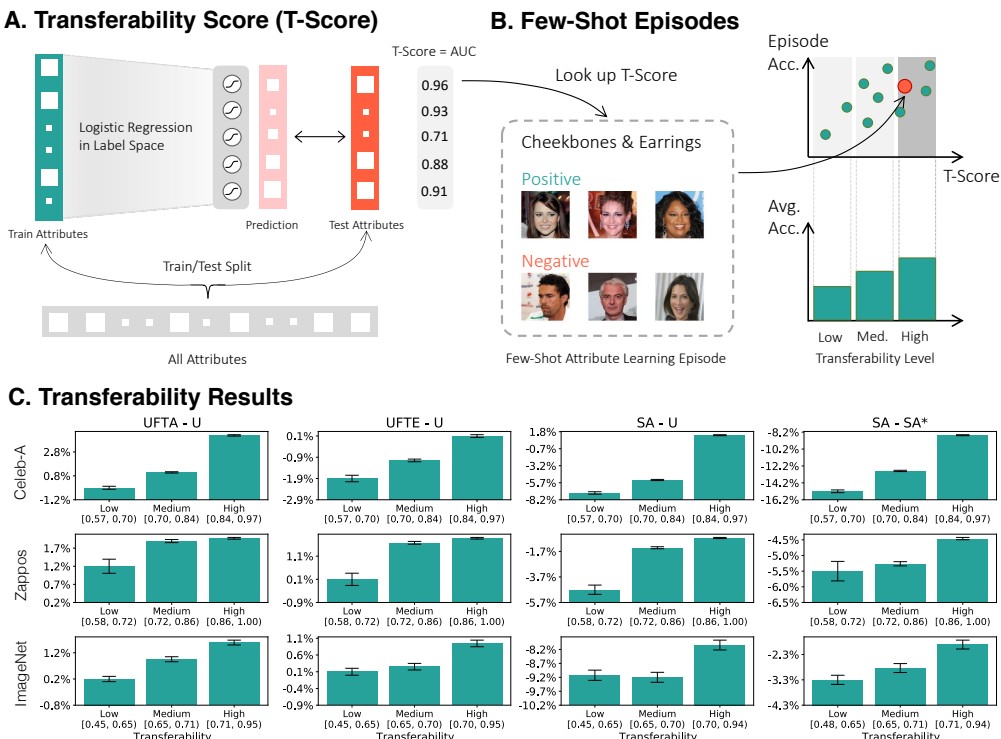

Figure 4: **Few-shot performance vs. transferability across training and test attributes. A:** Transferability score (T-score) is computed based on the AUC of a test attribute predicted by a logistic regression model on a set of training attributes. 100 different random splits across train/test attributes per split are used. **B:** Both episodic accuracy and T-scores are recorded on 60,000 episodes (600 episodes per split). Episodes are grouped into three bins in terms of their T-scores. **C:** Performance of training or finetuning on training attributes correlates with T-score. Error bars are standard errors in each bin.

importantly, we aim to predict the transferability between training and test splits by analyzing the training vs. test attributes. Each image has a complete attribute vector, describing the values of each attribute in the image. Some of these attributes are in the training set, and others in the test set. To quantify the transferability, we leverage the idea of mutual information. More concretely, we learn a logistic regression model that takes the training attribute vector in a particular image as input and predicts the value of one of the test attributes in that image. Each logistic regression model will generate an AUC score on held-out images, and we average them across the relevant test attributes in each episode, and we define this AUC score as the "transferability score." Our hypothesis is that more mutual information between the attribute label distributions will translate to higher transfer performance when the model is supervised on the training attributes and tested on the test attributes.

In Figure 4, we ran experiments using 100 random splits of training and test attributes. The results verify our hypothesis. We see positive correlation between the transfer performance and our transferability score: When subtracting U as a baseline, both UFTA and UFTE models get better when there is higher transferability (subtraction reduces the effect of per-episode variability). The same conclusion can be drawn when we subtract SA* from SA. By plotting the relation between U and SA, we show that supervised learning is more helpful when there is higher transferability in the label space whereas self-supervised learning is more flexible at adapting to novel target tasks.

## 6 CONCLUSION

Acquiring knowledge of new attributes is one of the most basic learning skills of an intelligent agent. In this paper, we propose few-shot attribute learning to enable this core functionality. We developed benchmarks using the Celeb-A, Zappos-50K, and ImageNet datasets to create learning episodes using existing attribute labels. This setting presents a strong generalization challenge, since the split in attribute space can make the training and test tasks less similar. Consequently standard supervised representation learning performs poorly on the test set, unlike recent benchmark results in few-shot learning of semantic classes. We found that unsupervised contrastive learning preserved more general features, and further finetuning yielded strong performance. We also discovered that the similarity in the attribute label space can roughly predict the gain obtained by supervised training and finetuning, which could provide some insight into the generalization behavior of representation learning algorithms when dealing with novel tasks at test time.

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

| Mean AUC | RND | PN | ID | SA | U | UFTE | UFTA | SA* |
|---|---|---|---|---|---|---|---|---|
| All (40) | 79.18 | 88.80 | 91.29 | 90.27 | 92.80 | **93.34** | **93.33** | 94.46 |
| Train+Test (27) | 82.27 | 93.38 | 94.31 | 94.23 | 95.78 | **96.53** | **96.52** | 97.18 |
| Train (14) | 84.40 | 96.04 | 95.34 | 96.04 | 96.43 | **97.23** | **97.23** | 97.50 |
| Test (13) | 79.96 | 90.52 | 93.19 | 92.63 | 95.08 | **95.78** | **95.76** | 96.84 |

Table 9: **Celeb-A attribute readout** performance of different representations, measured in mean AUC. RND denotes using a randomly initialized CNN; PN denotes ProtoNet.

| | SA | U | UFTE | UFTA | SA* | GT |
|---|---|---|---|---|---|---|
| LR | 77.4 | 79.2 | 82.2 | 83.1 | 87.1 | 95.8 |
| +L1 (1e-4) | 77.6 (+0.2) | 79.4 (+1.2) | 82.3 (+0.1) | 83.2 (+0.1) | 87.4 (+0.3) | 96.1 (+0.3) |
| +L1 (1e-3) | **78.2** (+0.8) | **80.2** (+1.0) | **82.4** (+0.2) | **83.8** (+0.7) | **88.4** (+1.3) | 97.1 (+1.3) |
| +L1 (1e-2) | 75.7 (−1.7) | 78.3 (−0.9) | 78.8 (−3.5) | 79.5 (−3.6) | 87.6 (+0.5) | **98.2** (+2.4) |

Table 10: **Effect of the L1 regularizer** on different representations for the validation set of Celeb-A 20-shot.

## A    ATTRIBUTE READOUT

In Tab. 9 and 11, we provide attribute readout performance with different learned representations. This is a similar task that measures the generalizability, but it does not evaluate the rapid learning aspect brought by few-shot learning.

## B    ABLATION STUDIES

Table 10 studies the effect of the L1 regularization. The benefit is especially noticeable on SA* and GT, since it allows the few-shot learner to have a sparse selection of disentangled feature dimensions.

## C    ADDITIONAL HEATMAP VISUALIZATION

We provide additional visualization results in Figure 5, 6, and 7, and we plot the heat map to visualize the LR classifier weights. Figure 5 includes SA*, U, and UFTE which are omitted in the main paper due to space limitations. Figure 6 and 7 visualize more information including both support and query examples in the episode, and some of the episodes are challenging to solve given just a few examples.

## D    ATTRIBUTE SPLITS OF CELEB-A

We include the attribute split for Celeb-A in Table 12. There are 14 attributes in training and 13 attributes in val/test. We discarded the rest of the 13 attributes in the original datasets since they are either hard to classify with the oracle classifier (e.g. big lips, oval face) or simply ambiguous (e.g. young, attractive).

## E    ATTRIBUTE SPLITS OF ZAPPOS-50K

The Zappos-50K dataset annotates images with different values relating to the following aspects of shoes: 'Category', 'Subcategory', 'HeelHeight', 'Insole', 'Closure', 'Gender', 'Material' and 'Toestyle'.

| Mean AUC | SA | SA* | U | UFTA |
|---|---|---|---|---|
| All (25 attributes) | 72.01 | 73.02 | 81.08 | **82.49** |
| Train+Test (21 attributes) | 73.43 | 78.98 | 80.14 | **82.37** |
| Train (11 attributes) | 72.69 | 75.86 | 80.63 | **82.43** |
| Test (10 attributes) | 72.01 | 74.98 | 81.08 | **83.30** |

Table 11: ImageNet-with-Attributes attribute readout binary prediction performance of different representations, measured in mean AUC.

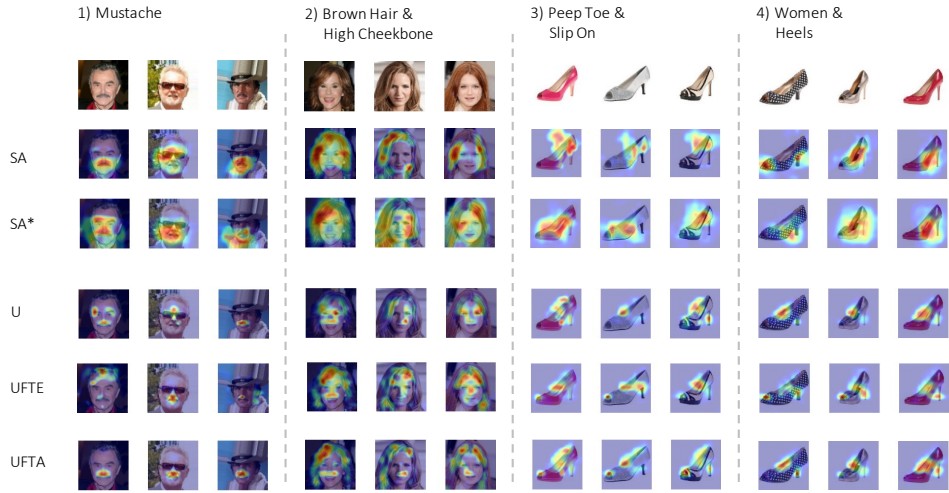

Figure 5: Additional visualization results, on 20-shot episodes, including more methods for comparison.

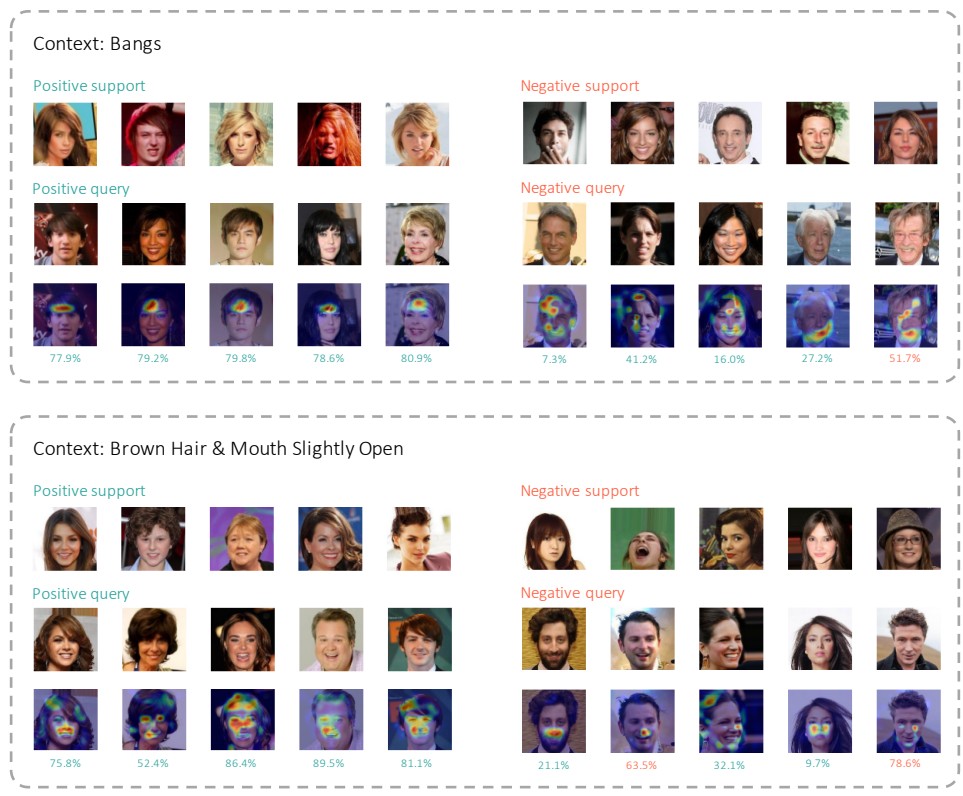

Figure 6: **Visualization of Celeb-A 20-shot LR classifiers using CAM on top of UFTA representations.** Context attributes that define the episode are shown above. Classifier sigmoid confidence scores are shown at the bottom. Red numbers denote wrong classification and green denote correct.

We discarded the 'Insole' values, since those refer to the inside part of the shoe which isn't visible in the images. We also discarded some 'Material' values that we deemed hard to recognize visually. We also modified the values of 'HeelHeight' which originally was different ranges of cm of the height of the heel of each shoe. Instead, we divided those values into only two groups: 'short heel' and 'high heel', to avoid having to perform very fine-grained heel height recognition which we deemed was too difficult.

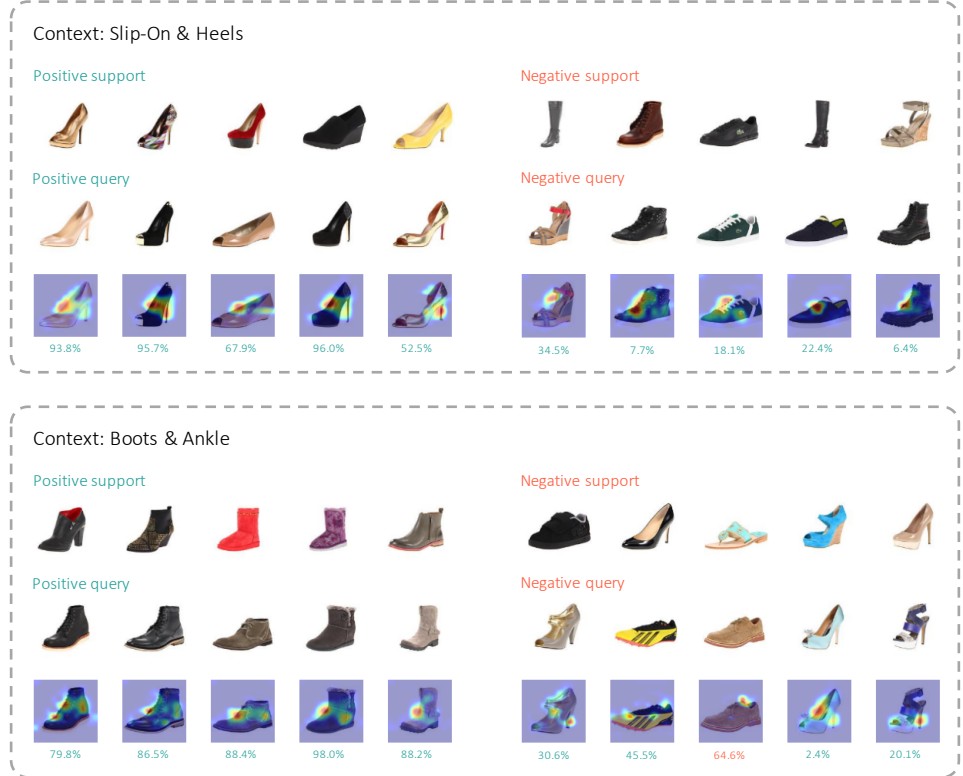

Figure 7: **Visualization of Zappos-50K 20-shot LR classifiers using CAM on top of UFTA representations.** Context attributes that define the episode are shown above. Classifier sigmoid confidence scores are shown at the bottom. Red numbers denote wrong classification and green denote correct.

| | | | |
|---|---|---|---|
| **Train** | 5_o_Clock_Shadow | Black_Hair | Blond_Hair | Chubby |
| | Double_Chin | Eyeglasses | Goatee | Gray_Hair |
| | Male | No_Beard | Pale_Skin | Receding_Hairline |
| | Rosy_Cheeks | Smiling | | |
| **Val/Test** | Bald | Bangs | Brown_Hair | Heavy_Makeup |
| | High_Cheekbones | Mouth_Slightly_Open | Mustache | Narrow_Eyes |
| | Sideburns | Wearing_Earrings | Wearing_Hat | Wearing_Lipstick |
| | Wearing_Necktie | | | |

Table 12: Attribute Splits for Celeb-A

These modifications leave us with a total of 79 values (across all higher-level categories). Not all images are tagged with a value from each category, while some are even tagged with more than one value from the same category (e.g. two different materials used in different parts of the shoe). We split these values into 40 'training attributes' and 39 'val/test attributes'.

We include the complete list of attributes in Table 13. The format we use is 'X-Y' where X stands for the category (e.g. 'Material') and Y stands for the value of that category (e.g. 'Wool'). We do this to avoid ambiguity, since it may happen that different categories have some value names in common, e.g. 'Short Heel' is a value of both 'SubCategory' and 'HeelHeight'.

## F    ATTRIBUTE SPLITS OF IMAGENET-WITH-ATTRIBUTES

We include the attribute split for ImageNet-with-Attributes in Table 14. There are 11 attributes in training and 10 attributes in val/test. We discarded the rest of the 4 attributes in the "shape" category (long, round, rectangular and square), since they are difficult to predict from the images.

| | | | | |
|---|---|---|---|---|
| **Train** | Category-Shoes
SubCategory-Boot
SubCategory-Slipper Heels
SubCategory-Over the Knee
Closure-Zipper
Closure-Snap
Gender-Boys
Material-Aluminum
Toestyle-Snub Toe
Toestyle-Almond | Category-Sandals
SubCategory-Slipper Flats
SubCategory-Athletic
HeelHeight-High heel
Closure-Elastic Gore
Closure-T-Strap
Material-Rubber
Material-Plastic
Toestyle-Bicycle Toe
Toestyle-Apron Toe | SubCategory-Oxfords
SubCategory-Short heel
SubCategory-Knee High
Closure-Pull-on
Closure-Sling Back
Closure-Spat Strap
Material-Wool
Toestyle-Capped Toe
Toestyle-Open Toe
Toestyle-Snip Toe | SubCategory-Heel
SubCategory-Flats
SubCategory-Crib Shoes
Closure-Ankle Strap
Closure-Toggle
Gender-Men
Material-Silk
Toestyle-Square Toe
Toestyle-Pointed Toe
Toestyle-Medallion |
| **Val/Test** | Category-Boots
SubCategory-Loafers
SubCategory-Heels
HeelHeight-Short heel
Closure-Slip-On
Closure-Button Loop
Gender-Girls
Material-Horse Hair
Toestyle-Moc Toe
Toestyle-Bump Toe | Category-Slippers
SubCategory-Boat Shoes
SubCategory-Prewalker
Closure-Lace up
Closure-Ankle Wrap
Closure-Monk Strap
Material-Suede
Material-Stingray
Toestyle-Wingtip
Toestyle-Wide Toe Box | SubCategory-Mid-Calf
SubCategory-Clogs and Mules
SubCategory-Prewalker Boots
Closure-Buckle
Closure-Bungee
Closure-Belt
Material-Snakeskin
Toestyle-Round Toe
Toestyle-Center Seam
Toestyle-Peep Toe | SubCategory-Ankle
SubCategory-Sneakers and Athletic Shoes
SubCategory-Firstwalker
Closure-Hook and Loop
Closure-Adjustable
Gender-Women
Material-Corduroy
Toestyle-Closed Toe
Toestyle-Algonquin |

Table 13: Attribute splits for Zappos-50K

| | | | | |
|---|---|---|---|---|
| **Train** | pink
shiny
metallic | spotted
rough
wooden | wet
striped
gray | blue
white |
| **Val/Test** | brown
orange
vegetation | green
yellow
smooth | violet
furry | red
black |

Table 14: Attribute Splits for ImageNet-with-Attributes

# G    FEW-SHOT ATTRIBUTE LEARNING TOY PROBLEM

In this section, we present a toy problem that illustrates the challenges introduced by the FSAL setting and the failures of existing approaches on this task. This simple problem captures the core elements of our FSAL tasks, including ambiguity, introducing novel attributes at test time, and the role of learning good representations. The primary limitation of this model is the fact that it is fully linear and the attribute values are independent—in a more realistic FSAL task recovering a good representation from the data is significantly more challenging, and the data points will have a more complex relationship with the attributes as in our benchmark datasets.

**Problem setup**    We define a FSAL problem where the data points $\mathbf{x} \in \mathbb{R}^m$ are generated from binary attribute strings, $\mathbf{z} \in \{0,1\}^d$, with $\mathbf{x} = A\mathbf{z} + \boldsymbol{\zeta}$ for some matrix $A \in \mathbb{R}^{m \times d}$ with full column rank and noise source $\boldsymbol{\zeta}$. Thus, each data point $\mathbf{x}$ is a sum of columns of $A$ with some additive noise.

In each episode, examples are labelled as positive when two designated entries of the attribute strings are both 1-valued, and negative otherwise. For the training episodes, the labels depend only on the first $d_1 < d$ entries of $\mathbf{z}$. At test time, the labels depend on the remaining $d - d_1$ attributes. The training and test episodes are generated by choosing two of the attributes in the respective sets. Then $k$ data points are sampled with positive labels (the two attributes are 1-valued) and $k$ with negative labels (at least one of the attributes is 0-valued).

**Linear prototypical network**    Now, consider training a prototypical network on this data with a linear embedding network, $g(\mathbf{x}) = W\mathbf{x}$. Within each episode, the prototypical network computes the prototypes for the positive and negative examples,

$$\mathbf{c}_j = \frac{1}{k} \sum_{\mathbf{x}_i \in S_j} g(\mathbf{x}_i) = \frac{1}{k} \sum_{\mathbf{x}_i \in S_j} \sum_{l=1}^{d} z_{il} W \mathbf{a}_l, \text{ for } j \in \{0, 1\},$$

where $S_j$ is the set of data points in the episode with label $j$, and $\mathbf{a}_l$ is the $l^{\text{th}}$ column of the matrix $A$. Further, the prototypical network likelihood is given by,

$$p(y = 0|\mathbf{x}) = \frac{\exp\left\{-\|W\mathbf{x} - \mathbf{c}_0\|_2^2\right\}}{\exp\left\{-\|W\mathbf{x} - \mathbf{c}_0\|_2^2\right\} + \exp\left\{-\|W\mathbf{x} - \mathbf{c}_1\|_2^2\right\}}.$$

The goal of the prototypical network is thus to learn weights $W$ that lead to small distances between data points in the same class and large distances otherwise. In the FSAL tasks, there is an addi-

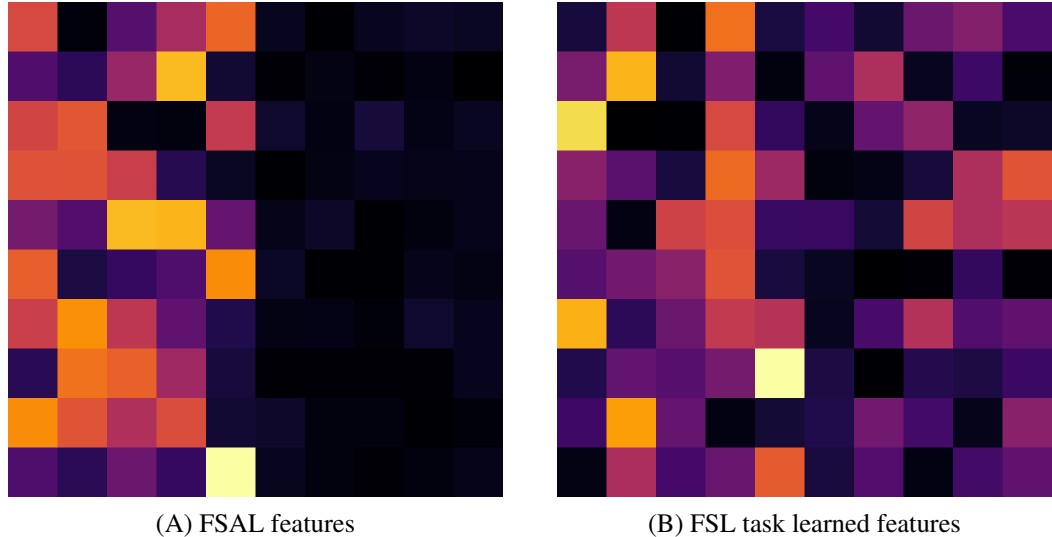

(A) FSAL features            (B) FSL task learned features

Figure 8: Projecting data features into prototypical network embedding space ($WA$) for the linear toy problem. Values closer to zero are darker in colour. On the FSAL task, the model destroys information from the test attributes to remove ambiguity at training time.

tional challenge in that class boundaries shift between episodes. The context (the choice of attribute entries) defining the boundary is unknown and must be inferred from the episode. However, with few shots (small $k$) there is ambiguity in the correct context — with a high probability that several possible contexts provide valid explanations for the observed data.

**Fitting the prototypical network**    Notice that under our generative model, with $\mathbf{x} = W\mathbf{z} + \boldsymbol{\zeta}$ and for $j \in \{0, 1\}$ we have,

$$W\mathbf{x} - \mathbf{c}_j = WA(\mathbf{z} - \frac{1}{k}\sum_{\mathbf{z}_i \in S_j} \mathbf{z}_i) + \frac{1}{k}\sum_i W\boldsymbol{\zeta}_i + W\boldsymbol{\zeta}.$$

Notice that if $\mathbf{v}_j(\mathbf{z}) = A(\mathbf{z} - \frac{1}{k}\sum_{\mathbf{z}_i \in S_j} \mathbf{z}_i) \in \text{Ker}(W)$, the kernel of $W$, then the entire first term is zero. Further, if $\mathbf{z} \in S_j$ (the same class as the prototype) then there is no contribution from the positive attribute features in this term. Otherwise, this term is guaranteed to have some contribution from the positive attribute features.

Therefore, if $W$ projects to the linear space spanned by the positive attribute features then $W\mathbf{v}_j(\mathbf{z})$ is zero when $\mathbf{z} \in S_j$ and non-zero otherwise. This means that the model will be able to solve the episode without contextual ambiguity. Then the optimal weights are those that project to the set of features used in the training set—destroying all information about the test attributes which would otherwise introduce ambiguity.

We observed this effect empirically in Figure 8, where we have plotted the matrix $\text{abs}(WA)$. Each column of these plots represents a column of $A$ mapped to the prototypical network's embedding space. The first 5 columns correspond to attributes used at training time, and the remaining 5 to those used at test time.

In the FSAL task described above, as our analysis suggests, the learned prototypical feature weights project out the features used at test time (the last 5 columns). As a result, the model achieved 100% training accuracy but only 51% test accuracy (chance is 50%).

We also compared against an equivalent problem set up that resembles the standard few-shot learning setting. In the FSL problem, the binary attribute strings may have only a single non-zero entry and each episode is a binary classification problem where the learner must distinguish between two classes. Now the vector $\mathbf{z}$ is a one-hot encoding and the comparison to the prototypes occurs only over a single feature column of $A$, thus there is no benefit to projecting out the test features. As

expected, the model we learned (Figure 8 B) is not forced to throw away test-time information and achieves 100% training accuracy and 99% test accuracy.

**Settings for Figure 8** We use 10 attributes, 5 of which are used for training and 5 for testing. We use a uniformly random sampled $A \in \mathbb{R}^{30 \times 10}$ and the prototypical network learns $W \in \mathbb{R}^{10 \times 30}$. We use additive Gaussian noise when sampling data points with a standard deviation of 0.1. The models are trained with the Adam optimizer using default settings over a total of 30000 random episodes, and evaluated on an additional 1000 test episodes. We used $k = 20$ to produce these plots, but found that the result was consistent over different shot counts.

