# OpenReview forum: "Few-Shot Attribute Learning"
_ICLR.cc/2022/Conference — ICLR 2022 Submitted_

### Official Review · Reviewer_ym8e · 2021-10-30

**Correctness:** 2
**Technical Novelty And Significance:** 2
**Empirical Novelty And Significance:** 2
**Recommendation:** 3
**Confidence:** 5

**Main Review:**

**Strengths

- Introduces a formulation for the problem of few-shot attribute learning.

- Proposes reasonable baselines and experimental results on three benchmarks.

- Provides some insight about learning behavior (explainability heatmaps, transferability of attribute, influence of shot number).


**Weaknesses

Basically I found that the article addresses an interesting issue about attributes as an intermediate representation for transferability but that it fails to provide convincing answers to this question mostly due to lack of rigor and precision in the way it is presented and justified. Most of my remarks go in this direction.

- The target scientific goal of the paper is unclear: Finding ways to better analyze transferability from attribute prediction? Justifying that supervised fine-tuning is needed for pre-training? Proposing a new algorithm to solve few-shot attribute learning?

- The task definition is also unclear: I didn't understand whether the attribute prediction task is only a binary classification on a single attribute (or a conjunction) from the unseen set, or a multiple-attribute detection problem? Is each episode considered as a "2-way/n shot" problem, the two classes being the target attribute present or not? The notations used to define such a task are ambiguous or incomplete.

- The real algorithmic novelty of the proposed approach is low: the main result seems to be that self-supervision yields better features for the problem of attribute detection (this is not a real surprise given the literature [1,2]) and that supervised fine-tuning also improves features. Indeed, the proposed approaches (UFTA & UFTE) are very elementary and should rather be considered as baselines.

- The transferability analysis could be interesting but is not clearly conducted. For instance, what does "subtracting U as a baseline" mathematically mean, how is it done? Also, it is said that the transferability score is an average on attributes in each episode, but I understood an episode as testing a single attribute (see my question about task definition). The formula used to compute it should be made more explicit as it is a major contribution of the paper.

- The evaluation protocol for FSAL only exploits one split on the attribute set: the study about transferability uses multiple random splits. Why not use such a protocol for the main task? It would give more consolidated results.

- Is attribute prediction fundamentally different from multi-label classification, an already studied area [3,4]? The introduction also speaks of expanding the vocabulary but the paper does not seem to evaluate such incremental capacity.

- Compared methods:  it is not clear how they are modified to take into account the attribute detection problem instead of a classification with exclusivity between classes. More details should be given, at least in the supplemental material.

- The idea of "context", presented as a key issue in the paper, is not clear: is it given by the training dataset or by the support set for each episode?

- It is now well established that self-supervised learning (SSL) is able to provide more general and robust features than supervision: the impact of the SSL method and of the DNN backbones however are important for classification (see [1]), and the proposed study only studies one configuration for each dataset. How really general are the results provided? The difference between the U algorithm (i.e. that skips the supervised fine-tuning part) and the two proposed baselines is often small: the role of the supervised fine-tuning is not obvious. Also, one could use pre-trained features learned using a bigger external dataset. If the main objective of the paper is to justify that a supervised fine-tuning part is useful - a reasonable assumption - more competing unsupervised features need to be compared.

- There are many other datasets with attributes or labels: COCO attribute, AWA, CUBS, Flower... which contain more attributes than the datasets used in the paper. Since the paper is to introduce a new problem, a justification of the benchmark is expected.

- A missing reference [5] with a clearer presentation of a  few-shot attribute learning problem.

- Writing style: Please avoid assertions such as "our proposed method produces *surprisingly* accurate and localized heatmaps that pinpoint the location of the attributes (e.g. mustache or cheekbone); this is impressive since no labeled information concerning these attributes was available during representation learning". It seems to imply that self-supervised representations magically locate attributes! The interesting scientific question would be to understand why such behavior happens, if it is a consequence of the self-supervision or supervised fine-tuning, for instance.

- Formal description: please describe with better precision the mathematical spaces underlying symbols, and their meaning: it helps to fully understand the paper.


[1] Sariyildiz, M. B., Kalantidis, Y., Larlus, D., & Alahari, K. (2021). Concept generalization in visual representation learning. In Proceedings of the IEEE/CVF International Conference on Computer Vision (pp. 9629-9639).

[2] Van Horn, G., Cole, E., Beery, S., Wilber, K., Belongie, S., & Mac Aodha, O. (2021). Benchmarking Representation Learning for Natural World Image Collections. In Proceedings of the IEEE/CVF Conference on Computer Vision and Pattern Recognition (pp. 12884-12893).

[3] Alfassy, A., Karlinsky, L., Aides, A., Shtok, J., Harary, S., Feris, R., ... & Bronstein, A. M. (2019). Laso: Label-set operations networks for multi-label few-shot learning. In Proceedings of the IEEE/CVF Conference on Computer Vision and Pattern Recognition (pp. 6548-6557).

[4] Li, Z., Mozer, M., & Whitehill, J. (2021). Compositional embeddings for multi-label one-shot learning. In Proceedings of the IEEE/CVF Winter Conference on Applications of Computer Vision (pp. 296-304).

[5] Xiang, L., Jin, X., Ding, G., Han, J., & Li, L. (2019) Incremental Few-Shot Learning for Pedestrian Attribute Recognition. IJCAI.


**Summary Of The Paper:**

The article proposes to address attribute detection with a small data constraint in the training phase (few-shot approach). The experimental protocol is divided in a series of episodes, each defined by a support set that specifies the attributes to be predicted and a query set used to evaluate the prediction. The generic algorithmic structure to solve this problem is divided in two steps: a common pre-training phase that combines self-supervised representation learning and supervised fine-tuning, followed by a supervised learning phase to solve each episode task. A training dataset containing image samples with attributes is used to learn the pre-trained representation space. The pre-training phase is compared to other approaches on three benchmarks implementing few-shot attribute prediction episodes. An indicator computed from a logistic regression between attribute values is proposed as a way to predict transferability between the learning dataset and the few-shot problem.


**Summary Of The Review:**


The question of addressing transferability at attribute level is interesting. A "few-shot" formulation to do so, as proposed in the article, is a possible direction.

However, the overall presentation of the problem is messy, superficial and lacks precision. The algorithm novelty is low and the experimental part relies on questionable protocols.

The article should be reworked with a clearer focus on the scientific hypothesis that is tested and better justified experimentations.

Final decision:

My conclusion about acceptance has not really evolved: the problem of attribute transferability addressed in the paper is interesting, but the way it is studied and presented must be improved and better justified (see detailed comment of rebuttal after author's response).

---

> ### Author Response · Authors · 2021-11-18
> **Authors' response to reviewer ym8e**
>
> > Basically I found that the article addresses an interesting issue about attributes as an intermediate representation for transferability but that it fails to provide convincing answers to this question mostly due to lack of rigor and precision in the way it is presented and justified. Most of my remarks go in this direction.
>
> We are glad to learn that the reviewer is an expert in our area and we would love to engage in a deeper discussion here.
>
> > The target scientific goal of the paper is unclear: Finding ways to better analyze transferability from attribute prediction? Justifying that supervised fine-tuning is needed for pre-training? Proposing a new algorithm to solve few-shot attribute learning?
>
> The main scientific goal of our paper is to propose a new few-shot learning task: few-shot attribute learning. This is an important contribution as studying this problem yields different insights. First, we show empirically that excelling in this task requires different modeling approaches compared to those that are currently popular for standard few-shot classification. Second, this novel task highlights the need to better understand the consequences of the choice of representation learning algorithm (e.g. self-supervised vs supervised, or combinations of these) which we believe is an understudied and important area of research. Finally, as we aim to analyze the results further, our transferability experiments provide a framework for understanding when these different representation learning approaches can be most successful.
>
> > The task definition is also unclear: I didn't understand whether the attribute prediction task is only a binary classification on a single attribute (or a conjunction) from the unseen set, or a multiple-attribute detection problem? Is each episode considered as a "2-way/n shot" problem, the two classes being the target attribute present or not? The notations used to define such a task are ambiguous or incomplete.
>
> Each episode is a 2-way/n-shot problem where one class is the positive class (examples that have one or a pair of attributes presented in the episode) and the other is the negative class (examples that don’t have that attribute). We believe that this is actually clearly described in Section 2, e.g. in Equation 1. Why is it only a binary classification problem? Since we are essentially asking the model to “detect” an attribute out of many potential attributes. And the problem of detection is inherently binary: either an attribute is there or not there. Using multi-way would not make sense for the attribute problem since the fact an image that has attribute A does not imply that it does not have attribute B, but on the other hand, multi-way implies an exclusive relationship. You could, however, construct multiple binary classification problems, but that’s just a slight variation of our setup by presenting multiple episodes at the same time.
>
>
> We do have two ways of defining attributes, either by taking conjunctions (binary attributes, e.g. ‘smiling and wearing hat’) or not (unary attributes, e.g. ‘smiling’). However, the task considered is always binary (i.e. 2-way) classification, for different definitions of the positive and negative class.
>
> Please let us know if this is clear and we are happy to further elaborate in the paper accordingly.
>
> > The real algorithmic novelty of the proposed approach is low: the main result seems to be that self-supervision yields better features for the problem of attribute detection (this is not a real surprise given the literature [1,2]) and that supervised fine-tuning also improves features. Indeed, the proposed approaches (UFTA & UFTE) are very elementary and should rather be considered as baselines.
>
> As we argued in the common response to all reviewers, the extent to which self-supervision leads to a performance boost was actually not expected. For instance, Tian et al. (2020) recently reported that self-supervision works roughly the same as pure supervision in few-shot classification (see Table 3 in that paper).  Regardless, we emphasize that our goal on the modeling front was to provide a strong baseline for studying our proposed few-shot attribute learning task, which is where the main novelty of this work lies. We agree that the individual components of our modeling pipeline aren’t novel, but we believe it is a strong empirical contribution to show that this particular combination (especially of self-supervised and supervised learning) can significantly surpass the performance of common baselines, and we expect this to incentivize future work to further study this finding. In line with this year’s ICLR review guidelines, we expect our work to be evaluated in terms of its empirical novelty, instead of its technical novelty.

---

> > ### Author Response · Authors · 2021-11-18
> > **part 2**
> >
> > > The transferability analysis could be interesting but is not clearly conducted. For instance, what does "subtracting U as a baseline" mathematically mean, how is it done? Also, it is said that the transferability score is an average on attributes in each episode, but I understood an episode as testing a single attribute (see my question about task definition). The formula used to compute it should be made more explicit as it is a major contribution of the paper.
> >
> > Thanks for the note, we will clarify in the paper too: we subtract the *performance* of U as a baseline (in terms of the AUC score). The average here is taken over *2000 test episodes* (each of which is a binary classification task).
> >
> > > The evaluation protocol for FSAL only exploits one split on the attribute set: the study about transferability uses multiple random splits. Why not use such a protocol for the main task? It would give more consolidated results.
> >
> > We agree that using more splits is always better for robustness. However, we emphasize that this leads to a very large number of experiments to run, and would burden future work to train such a large number of models against our benchmark tasks. Further, we would like to point out that the standard approach in few-shot classification is to only use one split, so our setup is consistent with common practice in that sense. In our transferability experiments, we produce several attribute splits to study transferability with reduced variance.
> >
> > > Is attribute prediction fundamentally different from multi-label classification, an already studied area [3,4]? The introduction also speaks of expanding the vocabulary but the paper does not seem to evaluate such incremental capacity.
> >
> > Studying learning new attributes is a required stepping stone for continually applying this skill. We view our work as setting the groundwork for incremental learning explorations of this task in the future, which is a very interesting direction.
> >
> > We agree that our task bears similarity to multi-label classification, and we will add these references to the related work section. However, a crucial difference is that our goal is to generalize to *previously unlabeled* attributes at test time, which is what leads to a challenging generalization problem. In contrast, [3] and [4] explored ways to perform set operations on multiple labels without concerning the underlying generalization issues.
> >
> > > Compared methods: it is not clear how they are modified to take into account the attribute detection problem instead of a classification with exclusivity between classes. More details should be given, at least in the supplemental material.
> >
> > All of the methods we used were designed for few-shot classification. Since our task can also be viewed as a 2-way classification problem, these are directly applicable without modification.
> >
> > > The idea of "context", presented as a key issue in the paper, is not clear: is it given by the training dataset or by the support set for each episode?
> >
> > The context is given by the support set for each episode. We thank the reviewer for bringing this up and we will clarify in the paper.

---

> > > ### Author Response · Authors · 2021-11-18
> > > **part 3**
> > >
> > > > It is now well established that self-supervised learning (SSL) is able to provide more general and robust features than supervision: the impact of the SSL method and of the DNN backbones however are important for classification (see [1]), and the proposed study only studies one configuration for each dataset. How really general are the results provided? The difference between the U algorithm (i.e. that skips the supervised fine-tuning part) and the two proposed baselines is often small: the role of the supervised fine-tuning is not obvious. Also, one could use pre-trained features learned using a bigger external dataset. If the main objective of the paper is to justify that a supervised fine-tuning part is useful - a reasonable assumption - more competing unsupervised features need to be compared.
> > >
> > > Regarding pre-trained features from a larger dataset: in our ImageNet-with-attributes experiments, indeed the self-supervised representations were trained on a larger dataset, and we show the positive effect of that on performance on our task. However, in our main experiments with Celeb-A and Zappos, we opted to train the self-supervised and supervised models on exactly the same data, to enable apples-to-apples comparisons between the resulting models which is important to our study.
> > >
> > > Indeed, the gain of supervised fine-tuning on top of U is sometimes small. Our transferability experiments were actually designed to investigate under which circumstances this is the case. We find empirically that the gain of supervised fine-tuning on top of U is largest when there is higher transferability (i.e. mutual information) between the training and test attributes, in line with our intuition. When the mutual information is low, relying on purely unsupervised features works the best.
> > >
> > > We agree that studying different architectures would also be useful for future work, but we argue that our experiments already are very extensive, involving several models and methods on multiple datasets. The fact that the conclusions are consistent across datasets is a good indication of the robustness of our findings.
> > >
> > > > There are many other datasets with attributes or labels: COCO attribute, AWA, CUBS, Flower... which contain more attributes than the datasets used in the paper. Since the paper is to introduce a new problem, a justification of the benchmark is expected.
> > >
> > > We have considered the datasets that the reviewer mentioned, but 1) COCO attribute focuses more on human object interaction attributes and these are more of an object detection problem rather than discovering intrinsic attributes of an object; 2) AWA and Flower focus on the appearance like colors and patterns and we felt that these attribute are too shallow and probably do not require a large amount of inference; 3) CUBS has adequate attributes, but unfortunately contains fewer than 10k images and we are left with 5k after the training/val/test split; this is not enough for self-supervised training.
> > >
> > > > A missing reference [5] with a clearer presentation of a few-shot attribute learning problem.
> > >
> > > We will include this reference. However, [5] only studied attribute learning in a restricted setting, pedestrian images, and we also do not believe that the incremental aspect is a better formulation of the problem since there are more fundamental generalization challenges in the space of novel attribute learning, which our paper aims to unveil.

---

> > > > ### Author Response · Authors · 2021-11-18
> > > > **part 4**
> > > >
> > > > > Writing style: Please avoid assertions such as "our proposed method produces surprisingly accurate and localized heatmaps that pinpoint the location of the attributes (e.g. mustache or cheekbone); this is impressive since no labeled information concerning these attributes was available during representation learning". It seems to imply that self-supervised representations magically locate attributes! The interesting scientific question would be to understand why such behavior happens, if it is a consequence of the self-supervision or supervised fine-tuning, for instance.
> > > >
> > > > We did hypothesize that it is the consequence of both self-supervision and supervised fine-tuning. We have provided our explanations on why SA or SA* failed to provide localized heatmap visualization. SA does not transfer well to novel attributes, whereas SA* encourages the attribute label information to spread across all spatial locations, which results in imprecise heatmaps. We believe that self-supervision leads to picking up more diverse features of the image, instead of focusing only on those that are discriminative of the training tasks. In Section 5.4, we further investigated the tradeoff between self-supervision and supervised fine-tuning. We found that supervised fine-tuning is more beneficial when there is more mutual information in the attribute label space between training attributes and test ones. We have never claimed or implied any phenomenons “magically” happen in the paper. The word “surprisingly”, which we are happy to revise, makes sense in the context in which it was used in the paper and was qualified by the subsequent clause that you quoted.
> > > >
> > > > > Formal description: please describe with better precision the mathematical spaces underlying symbols, and their meaning: it helps to fully understand the paper.
> > > >
> > > > We are not sure what the reviewer means by ‘mathematical spaces underlying symbols’. We would appreciate receiving direct pointers, but we will go over the paper and ensure that all equations are clearly explained.
> > > >
> > > > > However, the overall presentation of the problem is messy, superficial and lacks precision.
> > > > The algorithm novelty is low and the experimental part relies on questionable protocols.
> > > >
> > > > We thank the reviewer for their criticism, but we were surprised to see that the reviewer felt that we used questionable protocols (which we respectfully disagree with). Regarding novelty, as we wrote in the general comments, we argue that this paper achieves empirical novelty rather than technical novelty. We understand that our response probably won’t change the reviewer’s judgment in the end, but we firmly defend this work’s honor and scientific principles.

---

> > > > > ### Comment · Reviewer_ym8e · 2021-11-28
> > > > > **Comment on author's rebuttal**
> > > > >
> > > > >
> > > > > I thank the authors for their detailed answers. They clarified some of my concerns. However, there are still remaining issues for me even if "the work should be evaluated in terms of its empirical novelty, instead of its technical novelty" as suggested by the authors.
> > > > >
> > > > > "Although as some reviewers pointed out we should have run more self-supervised learners besides SimCLR"
> > > > >
> > > > > This is a important point I think, since one of your claim is that full supervision helps - adding new information should logically do that. However the gap between U and UFTE/A is small for at least two of the tested datasets, and it is likely that novel unsupervised methods will narrow it.
> > > > >
> > > > >
> > > > > "This suggests that the FSAL problem is more challenging than standard few-shot learning"
> > > > >
> > > > > I do not agree with this claim because the task studied in the paper is single and not multiple attribute prediction. This setting introduces an unnecessary constraint to the problem of attribute prediction.
> > > > >
> > > > > Given a set of attributes, not all combinations are observed in natural datasets, and many attributes are correlated (positively and negatively) or even redundant. This means that correlation between features could and should be exploited for prediction. Correlation can be used for transferability between old and novel attributes (what reviewer "tUmn" also questioned) but also between novel attributes themselves. The experimental setting proposed in the paper can't evaluate such property.
> > > > >
> > > > > Few-shot *single* attribute learning implies that attribute prediction should be made one by one: this is a simple but bad strategy for attribute prediction. Fundamentally, it misses the nature of attributes or labels as a *global*, compact and discriminating intermediate description of data usable for several tasks.
> > > > >
> > > > > Note that the paper uses multiple-attribute learning in the supervised loss of UFTA, meaning that the "normal" learning setting should make available multiple attribute annotations for each episode.
> > > > >
> > > > >
> > > > > "Questionable protocols"
> > > > >
> > > > > My concerns are about three points and have not changed:
> > > > >
> > > > > 1/ The nature of the datasets. You provided in the rebuttal some justification of your choice. Since your goal is to present a new problem, this justification should be in the text, with a review of candidates. There are many other sets with attributes or labels...
> > > > >
> > > > > 2/ The use of a single split for FSL. Your idea of exploiting multiple splits for the study of transferability was good and could have been extended to regular FSL evaluation to introduce new standards. Sometimes common practices need to be improved. Also, since one of the paper contributions is to experimentally study the tranferability of attributes, I would have expected a more rigorous protocol able to evaluate it, for instance using multiple splits of attributes.
> > > > >
> > > > > 3/ The reduction of the attribute description problem to single binary attribute detection (see comments above).
> > > > >
> > > > >
> > > > > "Formal description"
> > > > >
> > > > > Your symbols are not sufficiently defined in the text from a mathematical point of view (dimension, real/discrete, what they represent...): I needed such information to fully understand the paper.
> > > > >
> > > > >
> > > > > I am sorry to say that my conclusion about acceptance has not really evolved: the problem of attribute transferability addressed in the paper is interesting, but the way it is studied and presented must be improved and better justified.

---

### Official Review · Reviewer_tUmn · 2021-10-31

**Correctness:** 3
**Technical Novelty And Significance:** 2
**Empirical Novelty And Significance:** 2
**Recommendation:** 5
**Confidence:** 4

**Main Review:**

Strengths:
1. The problem setting, learning attributes in a few-shot setting, is novel. The problem may generalize to many realistic applications.
2. The authors reveal an interesting observation that self-supervised training can help the backbone to focus on generalizable features, which might also benefit the general FSL tasks.
3. The authors perform extensive experiments and ablation studies. The experiment results indicate that the proposed model leads to consistent improvement.

Weaknesses:
1. The three stages in the model are introduced in other papers before, which hinder the novelty of this paper. The self-supervised training is proposed in [1r], the fine-tuning stage adopts [2r], and the few-shot learning stage adopts logistic regression proposed in [3r].

2. Some model configuration and experiment details are not clear.
* If the LR outperforms NN and NC in the Few-shot learning stage, I'm curious about the performance of utilizing LR in the fine-tuning stage.

* The episode construction: on page 6 paragraph 1, the author randomly selects one or two attributes and looks for positive examples belonging to these attributes simultaneously. From Figure 2 we can observe that mustache is learned alone while Brown Hair and High Cheekbone are learned together. When would the author select two attributes and why is this happening?

*  In Table 3 the ablation study is performed under a 20-shot setting, however, nearly all methods achieve similar results using 20 samples. Why not use the results in the 5-shot setting?
*  In the CAM visualization of different methods, the author claim that the proposed method produces accurate and localized heatmaps for each attribute, which is even better than the oracle setting SA*. Why is the proposed model producing localized attention? I do not see any training constraints for this. More explanation for this phenomenon is appreciated.
* In related work the author claim that "In this paper, we study learning novel contextual similarities in the form of attributes using only a few training examples". What does context mean? And where is the evidence to support this claim?
3. My other major concern lies in Section 5.4, the generalization analysis. The author defines a transferability score and reveals a positive correlation between transferability and generalization ability. However, is this real transferability? My point is that some attributes are heavily correlated with each other the model might take advantage of the correlation to predict novel attributes. For instance, earings and lipsticks co-occur in most of the images. If the network has seen lipsticks as the base attribute, then given a few images with those two attributes showing together, the model will naturally misunderstand our intuition and predict the lipsticks instead of earings. Some more qualitative results might help us to understand if the network successfully generalizes to novel attributes, or just take advantage of the attribute correlation and misunderstand the attributes.

[1r] Ting Chen, Simon Kornblith, Mohammad Norouzi, and Geoffrey E. Hinton. A simple framework for contrastive learning of visual representations. CoRR, abs/2002.05709, 2020.

[2r] Jake Snell, Kevin Swersky, and Richard S. Zemel. Prototypical networks for few-shot learning. In Advances in Neural Information Processing Systems 30, NIPS, 2017.

[3r] Wei-Yu Chen, Yen-Cheng Liu, Zsolt Kira, Yu-Chiang Frank Wang, and Jia-Bin Huang. A closer look at few-shot classification. In Proceedings of the 7th International Conference on Learning Representations, ICLR, 2019.




**Summary Of The Paper:**

This paper investigates a new and interesting topic, few-shot attribute learning (FSAL), where the model is trained with base attributes with abundant samples and tested with novel attributes with only a few samples. Experiments were performed on three benchmark datasets with attribute annotation. The paper found that self-supervised learning helps to train a backbone that generalizes well to novel attributes, compared to training the backbone with only base attributes. The author also compares several few-shot learning models, e.g., MatchingNEt, MAML, ProtoNet, in this FSAL setting and found that the logistic regression layer works the best.

**Summary Of The Review:**

This paper introduces an interesting topic and performs extensive experiments to verify the conclusion. However, the technical novelty is hindered since most of the contributions are not new. Besides, there are some unclear points and claims that need to be clarified and discussed. I'm looking forward to the authors' responses.

---

> ### Author Response · Authors · 2021-11-18
> **Authors' response to reviewer tUmn**
>
> > The three stages in the model are introduced in other papers before, which hinder the novelty of this paper. The self-supervised training is proposed in [1r], the fine-tuning stage adopts [2r], and the few-shot learning stage adopts logistic regression proposed in [3r].
>
> We acknowledge that the three stages are separately introduced by other papers as the reviewer pointed out. However, our goal on the modeling front was to provide a strong baseline for studying our proposed few-shot attribute learning task, which is where the main novelty of this work lies. We agree that the individual components of our modeling pipeline aren’t novel, but we believe it is a strong empirical contribution to show that this particular combination (especially of self-supervised and supervised learning) can significantly surpass the performance of common baselines, and we expect this to incentivize future work to further study this finding. In line with this year’s ICLR review guidelines, we expect our work to be evaluated in terms of its empirical novelty, instead of its technical novelty.
>
> > Some model configuration and experiment details are not clear. If the LR outperforms NN and NC in the Few-shot learning stage, I'm curious about the performance of utilizing LR in the fine-tuning stage.
>
> Utilizing LR in the fine-tuning stage is akin to the MAML meta-learning algorithm, a variant of which we have already tried and reported in Table 2. In line with recent observations in few-shot classification, this variant underperforms simpler techniques (like SA) in our setup too.
>
> > The episode construction: on page 6 paragraph 1, the author randomly selects one or two attributes and looks for positive examples belonging to these attributes simultaneously. From Figure 2 we can observe that mustache is learned alone while Brown Hair and High Cheekbone are learned together. When would the author select two attributes and why is this happening?
>
> We sample a conjunction of attributes in order to form a class (e.g. ‘brown hair and high cheekbone’) since this allows us to significantly increase the total number of attributes at our disposal, both for training (if training episodically) as well as for testing episodes. For Celeb-A, we consider classes based on one or a pair of attributes, whereas for Zappos we focus on pairs of attributes. We will release all details of our task construction (including code) to enable future work to compare on our benchmarks.
>
> > In Table 3 the ablation study is performed under a 20-shot setting, however, nearly all methods achieve similar results using 20 samples. Why not use the results in the 5-shot setting?
>
> We used the 20-shot setting for ablation since in our study in Figure 3 we found that there is ambiguity in the task with smaller numbers of support examples and therefore we chose 20-shot in order to reduce the variance of the results. We will include 5-shot results in the appendix for future editions.
>
> > In the CAM visualization of different methods, the author claim that the proposed method produces accurate and localized heatmaps for each attribute, which is even better than the oracle setting SA*. Why is the proposed model producing localized attention? I do not see any training constraints for this. More explanation for this phenomenon is appreciated.
>
> This is a great point and we will elaborate on it further in the paper. This provides further evidence that self-supervision leads to learning representations that capture a variety of features, allowing a simple logistic regression approach downstream to pinpoint exactly the relevant ones for each test task. Our hypothesis why the proposed is better than SA* was provided in the manuscript: we hypothesized that the supervised objective of SA* encouraged the spread of the attribute information across spatial location, which results in less precise heatmap, whereas our model only fine-tuned briefly on the supervised objective.

---

> > ### Author Response · Authors · 2021-11-18
> > **part 2**
> >
> > > In related work the author claim that "In this paper, we study learning novel contextual similarities in the form of attributes using only a few training examples". What does context mean? And where is the evidence to support this claim?
> >
> > Our framework can be thought of as learning contextual similarity, due to the fact that images are grouped into classes in each episode in a way that is determined by the ‘context’ (i.e. the episode-specific definition of the positive class, determined by the episode’s selection of attribute(s)). For instance, an image of a blond smiling person and one of a blond frowning person may both be in the ‘positive class’ if it’s defined as being blond, whereas in another episode where the positive class is defined as smiling, one of these two images would be in the negative class. Therefore, another viewpoint for learning a new attribute in each episode is learning a form of contextual similarity, where the context must be inferred from the small support set to correctly label the query examples.
> >
> > > My other major concern lies in Section 5.4, the generalization analysis. The author defines a transferability score and reveals a positive correlation between transferability and generalization ability. However, is this real transferability? My point is that some attributes are heavily correlated with each other the model might take advantage of the correlation to predict novel attributes. For instance, earings and lipsticks co-occur in most of the images. If the network has seen lipsticks as the base attribute, then given a few images with those two attributes showing together, the model will naturally misunderstand our intuition and predict the lipsticks instead of earings. Some more qualitative results might help us to understand if the network successfully generalizes to novel attributes, or just take advantage of the attribute correlation and misunderstand the attributes.
> >
> > This is entirely in line with our intuition of transferability. Formally, if there is larger mutual information between the training and test attributes (as in your earrings and lipstick example) we say that the transferability is higher (i.e. it is easier for a model to predict the test attribute labels given the training attribute labels). We did also provide qualitative results in the heatmap visualization and our model seems to provide sensible groundings of the attributes.

---

> ### Comment · Reviewer_tUmn · 2021-11-29
> **Final decision**
>
> This paper introduces an interesting topic, however, there are some unclear statements, and some key experiments and analyses are still missing.
> After reading other reviewers' comments and the author's responses carefully, I'm leaning to keep the score unchanged.

---

### Official Review · Reviewer_5A2w · 2021-11-03

**Correctness:** 4
**Technical Novelty And Significance:** 2
**Empirical Novelty And Significance:** 3
**Recommendation:** 5
**Confidence:** 4

**Main Review:**

*** Positive aspects:
- The paper is clear, and well written. The contributions are also clear.
- The proposed approach delivers good performance against a set of reasonable few-shot learning baselines. More generally, all aspects of the experiments seem good. In particular, I appreciate the effort put into ensuring that no outside data is used in part 1 (which is a problem often acknowledged in zero-shot learning) [1]
- The problem setting is clearly motivated.
- Ablation studies are performed, giving more details into the effect of the different components.

*** Concerns:
There are no major concerns with the approach: the proposed method seems to work well.

Among the less important concerns, the most important one is that overall the proposed method is an assembly of previously trialed approaches. Combining unsupervised learning techniques with classical approaches to few-shot adaptation does not constitute a novel method.  In this review, I am not arguing for novelty for novelty's sake. I am rather just mentioning that the paper consists in introducing a variation on a classical learning problem (few shot strategies with attributes have been extensively studied in the literature), which is then solved using techniques that were designed for related problems.

*** Questions
- What motivated the choice of few-shot algorithm in Part 3 of the learning algorithm? Was there a theoretical motivation, or rather a practical one?
- How are hyper-parameters for each of the 3 parts optimized? All together or separately for each?
- What would be the effect of injecting attribute information directly in Part 1 (by an auxiliary loss e.g., which would combine part 1 and 2)?
- I wanted to confirm the following with the authors. When running experiments on Imagenet-1K, the SimCLR model they use has never seen any of the the test images, is that correct?
- Have the authors tried other self-supervised learning algorithms for part 1? Did other methods lead to additional insights, apart from performing worse? (i.e. is there a link between different families of self-supervised learning algorithms, and performance?).


*** References
[1] Xian, Yongqin, et al. "Zero-shot learning—a comprehensive evaluation of the good, the bad and the ugly." IEEE transactions on pattern analysis and machine intelligence 41.9 (2018): 2251-2265.

**Summary Of The Paper:**

The authors consider the problem of few-shot attribute learning. Contrary to most past approaches on learning with attributes, including classical zero-shot learning, the authors deal with the case where the attributes seen at test-time are previously unseen. As a result, few samples must be used to quickly adapt the learning algorithm to new attributes.
The approach works in 3 stages. The first uses a classical representation learning algorithm to learn a visual representation, SimCLR. Following this, the network is given supervised training to fine-tune the representation. In the last step, the authors adapt a previously devised few-shot learning strategy that consists in training a linear classifier.


**Summary Of The Review:**

The approach works, the components are well motivated. Despite this, in my opinion, the approach can be understood as a variation on more classical attribute-related few-shot learning tasks, solved by combining methods that work well on these. I would like to highlight that, despite the previous point, I feel the proposed idea has scientific merit, hence borderline score I have given currently.

---

> ### Author Response · Authors · 2021-11-18
> **Authors' response to reviewer 5A2w**
>
> > What motivated the choice of few-shot algorithm in Part 3 of the learning algorithm? Was there a theoretical motivation, or rather a practical one?
>
> Since the main novelty of our work lies in defining an interesting new formulation of few-shot learning, we decided to experiment with a wide range of commonly-used models in the field and attempted to provide a strong baseline for future work. We are particularly interested in understanding the role of self-supervised versus supervised representation learning for this problem. Given our intriguing observations that, in contrast to standard few-shot classification, self-supervision brings a significant boost in performance, we also investigated ways of combining the two. Our transferability analysis further studies when such a combination is most beneficial. We expect that our findings will lead future authors to theoretically investigate our new setting too.
>
> > How are hyper-parameters for each of the 3 parts optimized? All together or separately for each?
>
> Unsupervised pre-training can be time consuming and therefore we re-used many of the default hyperparameters from SimCLR directly. Other hyperparameters are optimized separately for each stage.
>
> > What would be the effect of injecting attribute information directly in Part 1 (by an auxiliary loss e.g., which would combine part 1 and 2)?
>
> We did try this and our preliminary results in that direction were less promising than the sequential framework we adopted. One reason is that the supervised loss decreases much faster than the self-supervised loss, and so it is hard to balance the two objectives. Moreover, we emphasize that the purpose of this approach is to provide a strong baseline on our new formulation and to incentivize other design choices to be studied in the future. We are not claiming that this is necessarily the optimal methodology.
>
> > I wanted to confirm the following with the authors. When running experiments on Imagenet-1K, the SimCLR model they use has never seen any of the the test images, is that correct?
>
> The SimCLR model we used was an off-the-shelf model pre-trained on ImageNet, so it has probably seen the test images. This is one of the reasons that we report these experiments separately compared to those for our other datasets. However, the fact that this would lead to a model that can learn unlabeled attributes from few examples at test time was not obvious and still constitutes an interesting finding.
>
> > Have the authors tried other self-supervised learning algorithms for part 1? Did other methods lead to additional insights, apart from performing worse? (i.e. is there a link between different families of self-supervised learning algorithms, and performance?).
>
> We did not. We opted to focus on SimCLR due to being a simple and popular algorithm that was empirically reported to perform very strongly, but we agree that this is a useful avenue for future work, and we think that further performance boosts may be observed by utilizing more recent self-supervised methods. However, a comparison of different self-supervised algorithms was not the objective of our work.

---

> > ### Comment · Reviewer_5A2w · 2021-12-06
> > **Final decision**
> >
> > I thank the authors for their response and acknowledge their efforts in responding. After the author's response, most of my initial concerns remain:
> > - The machine learning problem considered in the paper is novel by itself but is addressed by a direct combination of pre-existing methods. I am a bit concerned about the strengths of the conclusions as (1) there is possible contamination of the train set with test images (as mentioned by the authors, a problem which in the past was deemed important enough to warrant a redefinition of train/test splits in classical zero-shot learning [1]) and (2) the hyperparameters have not been re-optimized.
> > - Few alternative methods have been tested. In each step of the proposed solution, the authors have taken the most recent/strongest approach without verifying what other methods could be substituted for it (e.g. choice of SimCLR), or studying the impact the choice of a given method has on the result.
> >
> > As noted previously, the idea has scientific merit despite this. As my assessment of the paper has not changed, I will be maintaining my score.
> >
> > ----
> > [1] Xian, Y., Schiele, B., & Akata, Z. (2017). Zero-shot learning-the good, the bad and the ugly. In Proceedings of the IEEE Conference on Computer Vision and Pattern Recognition (pp. 4582-4591).

---

### Official Review · Reviewer_7SAf · 2021-11-04

**Correctness:** 3
**Technical Novelty And Significance:** 3
**Empirical Novelty And Significance:** 3
**Recommendation:** 6
**Confidence:** 3

**Main Review:**

## Setting (FSAL)
The problem of rapid learning of attributes is well-motivated and practical. At a high level, Section 4 put the proposed setting in different contexts well.

## Benchmark
While the three proposed datasets are diverse in domains (faces, shoes, and general), I feel, however, that there is not enough concrete discussion about comparisons between proposed benchmarks and existing ones: FC100, tiered-ImageNet, Meta-Dataset, as well as Arnold and Sha 2021. My concern is that the proposed benchmarks seem to be very limited in the number of attributes (~20-80 in total each). Further, random splits are adopted without leveraging insights from previous work that split the semantic object classes. Would it be possible to put your task construction in the context of your goal of understanding generalization as well as in the context of related work that split the data in different ways?

## Experiments
Experiments are quite extensive with multiple baselines considered, with the proposed approaches improving upon existing ones further. However, it is unclear how the findings in Section 5.3 differ from previous findings when semantic object classes are considered. What would be our takeaway from the trend that SA performs poorly and U performs well?

## More detailed comments
- Setting: The claim that the setting with attributes is different from the standard few-shot setting on object classes is sound. I agree that the setting could be useful in understanding “the generalization performance semantic classes in standard few-shot learning” [Last sentence of page 1, last sentence of the intro]. However, I was wondering if the authors could say something more specific about what kind of insights we expect from exploring the setting? For example, how exactly would this offer “a more systematic framework to measure the relatedness and transferability between the traing and test set” [Last sentence of “Few-shot learning” in Section 4)? Do you expect this setting to be harder or easier than the one with object classes? Do you expect that the reuse of images for learning different attribute concepts in different stages might make the problem easier?
- Related Work: Should Arnold and Sha 2021 fall into “Few-shot learning” instead of “Generalization to novel tasks”. More generally, perhaps more discussion on how you separate these two items.
- Experiments: Have you tried “Baseline++” (Chen et al. 2019) for SA?



**Summary Of The Paper:**

This paper proposes the problem of few-shot attribute learning (FSAL) that follows the standard few-shot/meta-learning paradigm but focuses on attributes instead of object classes. The authors argue in Section 2 that the multi-label nature of attributes (smiling & wearing eyeglasses can be present simultaneously) makes this problem different; the context of each positive example in each episode can drastically change what the model is asked to learn.

This paper also proposes three benchmark datasets to study this problem, based on Celeb-A, Zappos50K, and ImageNet-with-Attributes. The paper evaluates multiple existing approaches on these benchmarks as well as improves upon them further.


**Summary Of The Review:**

Novel and potentially useful setting to study but some concerns on execution related to benchmarks and how we should interpret the results.

---

> ### Author Response · Authors · 2021-11-18
> **Authors' response to reviewer 7SAf**
>
> We thank the reviewer for the detailed and insightful comments.
>
> > There is not enough concrete discussion about comparisons between proposed benchmarks and existing ones: FC100, tiered-ImageNet, Meta-Dataset, as well as Arnold and Sha 2021. My concern is that the proposed benchmarks seem to be very limited in the number of attributes (~20-80 in total each).
>
> If the goal is to learn novel attributes at test time, we could not think of a way to leverage the existing few-shot learning datasets such as FC100, TieredImageNet, MetaDataset, and ArnoldSha21 since there are no attribute annotations in these datasets. Although the number of attributes is not as large in our datasets, the fact that we defined classes based on conjunctions of binary attributes produces a large label space. More generally, even with a number of attributes in the tens, a rich set of classes can be defined based on combinations of these attributes.
>
> > Further, random splits are adopted without leveraging insights from previous work that split the semantic object classes. Would it be possible to put your task construction in the context of your goal of understanding generalization as well as in the context of related work that split the data in different ways?
>
> We summarize two concurrent works that aim to understand generalization in the context of semantic object classes, one of which is kindly provided by one of the other reviewers. Arnold & Sha (2021) split the data in terms of the clustering distances in the embedding space of a pre-trained network. Sariyildiz et al. (2021) use the WordNet hierarchy to compute semantic distances. In our paper, we instead split the data in the attribute space, and if we assume that semantic classes are combinations of attributes, then a disjoint attribute split will imply further semantic distances. In other words, while Arnold & Sha rely on a pretrained network, we look into the attribute space for traces of transferability. One of the highlights of our generalization experiment in Section 5.4 is that our transferability score is data-driven and input independent, as it only depends on the similarity in the label space of attributes. In contrast, the pretrained network in Arnold & Sha (2021) can be sensitive to changes in the training images, whereas Sariyildiz et al. (2021)’s use of WordNet hierarchy relies on a manually specified semantic graph and therefore is not data-driven.
>
> > However, it is unclear how the findings in Section 5.3 differ from previous findings when semantic object classes are considered. What would be our takeaway from the trend that SA performs poorly and U performs well?
>
> We are not sure which work on semantic object class learning the reviewer is specifically referring to here. In contrast to standard few-shot learning, where supervised pre-training is typically a strong baseline, here we found that supervised pre-training (i.e. SA) performs poorly. We further found that self-supervised pre-training is essential for a generalizable representation for our task. In Section 5.3 we also investigated the task ambiguity of our task, since it requires more support examples in order to be certain of the semantic meaning of the support set (Figure 3). We made an attempt to understand why in our settings self-supervised pre-training generally out-performs supervised pre-training and yet, at the same time, supervised fine-tuning is beneficial. We evaluated the transferability score in the attribute space and we found higher transferability leads to more improvement using supervised learning.
>
> Lastly, since our main goal is few-shot attribute learning, and we have laid important groundwork for this goal, casting all of our conclusions only in terms of how they relate to semantic object classification is not a fair evaluation of our contributions.
>
> > However, I was wondering if the authors could say something more specific about what kind of insights we expect from exploring the setting? For example, how exactly would this offer “a more systematic framework to measure the relatedness and transferability between the traing and test set” [Last sentence of “Few-shot learning” in Section 4)?
>
> As we explained in the previous question, our attribute splits provide a controllable way to define transferability between training and testing environments. The attribute-based transferability score is data-driven and input independent, and both of which are merits of the proposed analytical framework. Again, these insights are generated through the lens of few-shot attribute learning, and we are not creating these splits just for the sake of understanding generalization of semantic object classes.

---

> > ### Author Response · Authors · 2021-11-18
> > **part 2**
> >
> >
> >
> > > Do you expect this setting to be harder or easier than the one with object classes?
> >
> > We believe that the FSAL setting is harder than standard few-shot learning of randomly splitting object classes. Further investigation of the generalization challenge presented in standard few-shot classification vs. FSAL would make for interesting future work. In the meantime, our empirical observation that standard pre-training does poorly on our setup supports this hypothesis.
> >
> > > Do you expect that the reuse of images for learning different attribute concepts in different stages might make the problem easier?
> >
> > We did not reuse the images for learning different attribute concepts in different stages. In fact, we made sure that there is no overlap between the images presented at training and test time! During the training phase though, if training episodically, the same image may play different roles in different episodes. For example, an image of someone smiling would be part of the positive class in one episode, if the positive class is 'smiling', but it may be part of the negative class in another episode, if in that episode the positive class is defined as 'wearing a hat'. We believe that this in fact increases the complexity of the problem. Please refer to Appendix G for our analysis.
> >
> > > Experiments: Have you tried “Baseline++” (Chen et al. 2019) for SA?
> >
> > Our LR classifier uses normalized features and it can be viewed as having a similar setup of Baseline++, which uses both normalized features and weight vectors.

---

### Author Response · Authors · 2021-11-18
**Authors' general response**

We thank all reviewers for their valuable time and insightful comments. We make a few general remarks here and respond to individual comments to each review below.

**Overall contributions:** We believe that our work on few-shot attribute learning (FSAL) not only lays the groundwork that allows machine learning algorithms to acquire new attribute vocabularies at test time, but also provides new insight into the generalization behavior of few-shot learners. One surprising finding from our experiments is that supervised learners and meta-learners do not transfer well to novel attributes, since their training objectives make them only sensitive to the set of training attributes. This is in contrast to standard few-shot classification, where pretraining on a set of training classes provides strong transfer performance to test classes. This suggests that the FSAL problem is more challenging than standard few-shot learning, and we have devoted a relatively large proportion of the paper trying to unveil some of the mysteries why supervised and meta learners fail to transfer well (See Section 5.4 and Appendix G).

**Technical vs. empirical novelty:** In designing the models, we opted for a simpler design, since the main contribution of this work lies in its novelty as a new few-shot learning task and our experimental framework and investigations of novel attribute generalization, rather than its technical novelty of a new model. When few-shot learning was first defined a few years ago, many papers at the time focused on developing new models for this task, but later people found out that simply using pre-trained representations can get equal if not better performance on few-shot learning. Therefore, based on the lesson learned, we believe that proposing a set of strong baselines for a new task is at least as beneficial as new models to the community in the long run. Therefore, in line with the spirit of this year’s ICLR review guideline, we argue that our work should be evaluated in terms of its empirical novelty, instead of its technical novelty.

**The relation between self-supervised learning and few-shot generalization:** The fact that standard supervised learners and meta-learners show poor performance on novel attribute learning inspired us to investigate whether unsupervised pre-training can perform well on the task, and preserve information that is useful for learning new attributes at test time. One reviewer says that it is “well established” that self-supervised learners perform better on transfer learning tasks. While this may be true for transfer learning, it certainly has not been well established for standard few-shot classification. Incorporating self-supervision has been studied before in few-shot learning, but the gains reported are not impressive. In fact, Tian et al (2020) reported that ‘representations learned with state-of-the-art self-supervised methods achieve slightly worse performance than fully supervised methods’. This observation is in stark contrast with our results. In particular, we hypothesized that the large performance boost we observed from self-supervision is due to the stronger form of test-time generalization required for our new setting, and we are the first to study how this difference in formulation impacts the performance of different methods. Although as some reviewers pointed out we should have run more self-supervised learners besides SimCLR, a criticism that we would agree with, our findings are not purely out of luck as we verified the hypothesis across three very different datasets (Celeb-A, Zappos, and ImageNet with Attributes). The large boost of self-supervision that we observe is an important empirical finding that we trust will lead to interesting follow up work to further investigate the role of self-supervision and when it is most beneficial.

---

> ### Author Response · Authors · 2021-11-18
> **Authors' general response (part 2)**
>
>
>
> **Value of supervised fine-tuning:** Our best model does not simply do self-supervised pre-training; we also find that a small amount of supervised fine-tuning on the training set also leads to performance improvements. This finding is intriguing: if supervised learning alone fails badly, why would self-supervised learning followed by supervised fine-tuning be any better? In the self-supervised learning literature, fine-tuning was only found to be helpful for semi-supervised classification of the training classes (Chen et al., 2020). As one reviewer points out that there are correlations between attribute labels and perhaps the correlation plays a role in terms of how much improvement supervised pre-training or fine-tuning can lead to. Indeed, we investigated this problem more deeply in our transferability experiment in Section 5.4 of our paper. Essentially, we aim to answer the question of, is it better to perform self-supervised learning vs. supervised learning in a transfer learning setting. Formally, we hypothesize that high mutual information in the label space can lead to higher transfer learning performance for supervised learning, whereas if there is less correlation in the label space, one should instead consider just using self-supervised pre-training. This is a natural hypothesis, and we found this hypothesis holds up strongly on all three datasets. Prior works investigated the transferability of few-shot learners through a pre-trained embedding space (e.g. Arnold and Sha (2021)), whereas we consider our transferability analysis in the label space. Our conclusion on supervised pre-training and fine-tuning could lead to newer insights for transfer learning and few-shot learning in general.
>
> **Summary:** We developed a new set of experiments for the task of few-shot attribute learning, and not only is it a cornerstone towards continual and flexible learning of new semantic classes, it also leads to more systematic understanding of the limitation of supervised transfer learning for this new type of test-time generalization. We hope that, through the lenses of few-shot attribute learning, the findings of this paper could open up the dialogue towards more practical understanding on generalization of novel concepts.
>
> **References:**
> - Tian et al. Rethinking Few-Shot Image Classification: a Good Embedding Is All You Need? ECCV’20
> - Chen et al. Big Self-Supervised Models are Strong Semi-Supervised Learners. NeurIPS’20.
> - Arnold and Sha. Embedding Adaptation is Still Needed for Few-Shot Learning. 2021.

---

### Author Response · Authors · 2021-11-23
**Rebuttal revision**

We have updated our manuscript. To summarize, we made the following changes:
* Added a figure that explains the setup of the transferability experiment on page 9.
* Added and discussed the multi-label and attribute few-shot learning references in the related work section.

-Authors

---

### Author Response · Authors · 2021-11-26
**Please respond during discussion period**

Dear Reviewers,

We kindly remind you that there are only a couple of days remaining in the discussion period. We went to significant effort to provide detailed responses to your reviews. We answered your specific questions and clarified important points around the contributions of our work. We are keen to discuss this with each of you and hope that you will take the time to respond.

Thank you,
Authors of "Few-Shot Attribute Learning"

---

### Decision · Program_Chairs · 2022-01-20

**Decision:**

Reject

**Comment:**

This paper has been reviewed with four expert reviewers. The reviewers have reached the consensus that the paper is not yet ready for publication. The main concerns are related to novelty. All reviewers gave substantial and constructive feedback. Following the recommendation of the reviewers, the meta reviewer recommends rejection.